# MOG analogues to explore the MCT2 pharmacophore, α-ketoglutarate biology and cellular effects of N-oxalylglycine

Louise Fets [1,4], Natalie Bevan[1,5], Patrícia M. Nunes[1,5], Sebastien Campos[2], Mariana Silva dos Santos [3], Emma Sherriff[2], James I. MacRae [3], David House[2] & Dimitrios Anastasiou [1✉]

α-ketoglutarate (αKG) is a central metabolic node with a broad influence on cellular physiology. The αKG analogue N-oxalylglycine (NOG) and its membrane-permeable pro-drug derivative dimethyl-oxalylglycine (DMOG) have been extensively used as tools to study prolyl hydroxylases (PHDs) and other αKG-dependent processes. In cell culture media, DMOG is rapidly converted to MOG, which enters cells through monocarboxylate transporter MCT2, leading to intracellular NOG concentrations that are sufficiently high to inhibit glutaminolysis enzymes and cause cytotoxicity. Therefore, the degree of (D)MOG instability together with MCT2 expression levels determine the intracellular targets NOG engages with and, ultimately, its effects on cell viability. Here we designed and characterised a series of MOG analogues with the aims of improving compound stability and exploring the functional requirements for interaction with MCT2, a relatively understudied member of the SLC16 family. We report MOG analogues that maintain ability to enter cells via MCT2, and identify compounds that do not inhibit glutaminolysis or cause cytotoxicity but can still inhibit PHDs. We use these analogues to show that, under our experimental conditions, glutaminolysis-induced activation of mTORC1 can be uncoupled from PHD activity. Therefore, these new compounds can help deconvolute cellular effects that result from the polypharmacological action of NOG.

[1] Cancer Metabolism Laboratory, The Francis Crick Institute, London, UK. [2] Crick–GSK Biomedical LinkLabs, London, UK. [3] Metabolomics Science Technology Platform, The Francis Crick Institute, London, UK. [4]Present address: Drug Transport and Tumour Metabolism Lab, MRC London Institute of Medical Sciences, London, UK. [5]These authors contributed equally: Natalie Bevan, Patrícia M. Nunes. ✉email: dimitrios.anastasiou@crick.ac.uk

The study of metabolism has long been aided by the use of metabolite analogues that allow the rapid and reversible inhibition of enzymes and pathways in different experimental settings[1]. In the field of cancer metabolism, analogue compounds such as 2-deoxyglucose, 6-diazo-5-oxo-L-norleucine (DON) and dichloroacetate (DCA) continue to complement genetic approaches to dissect the strengths and vulnerabilities associated with oncogene-driven metabolic changes in tumours[2–4]. Metabolite analogues are also among some of the most important clinically used chemotherapeutic compounds: from gemcitabine and 5-fluorouracil (5-FU), nucleoside analogues used as therapies in pancreatic and colorectal cancer; to methotrexate and pemetrexed, folate analogues administered to treat a range of malignancies[5,6]. The development and refinement of metabolite analogues can therefore provide valuable tools for mechanistic studies of both metabolism and tumorigenesis.

α-ketoglutarate (αKG) is a key metabolic node and understanding its complex biology has been facilitated by the structural analogue *N*-oxalylglycine (NOG), which has been extensively used in vitro along with its cell-permeable derivative, dimethyloxalylglycine (DMOG)[7–9] (Fig. 1a). Most commonly, DMOG is utilised to elicit hypoxia signalling by inhibiting prolyl hydroxylase domain (PHD) enzymes leading to stabilisation of the transcription factor Hypoxia Inducible Factor 1α (HIF1α)[8,10]. HIF1α stabilisation is a therapeutic aim in conditions ranging from ischaemia and anaemia to inflammatory diseases[11,12], and, in these settings, previous studies have used DMOG to demonstrate the potential therapeutic benefits of inhibiting PHDs[13,14].

As well as being a cofactor for αKG-dependent enzymes, αKG is also the entry point for glutamine carbons into the TCA cycle, a substrate for a large number of transaminase reactions, and has also been shown to mediate the activation of the mechanistic target of rapamycin complex 1 (mTORC1) by glutamine[15]. αKG can modify epigenetic profiles during development[9] and in pathogenic contexts[16] by regulating ten-eleven translocation (TET) hydroxylases and Jumonji demethylases. Additionally, αKG can influence aging[17], through an as-yet unclear mechanism, highlighting that there is still much to be discovered about the physiological functions of this metabolite.

Though DMOG is able to inhibit PHDs and thereby stabilise HIF1α in a broad range of cell lines, it is selectively toxic to some, in a manner that strongly correlates with the expression level of the monocarboxylate transporter MCT2[18], a member of the SLC16 family of transporters. DMOG is unstable in cell culture media and is non-enzymatically converted to the monocarboxylate methyl-oxalylglycine (MOG) with a half-life of 10 min. MOG is a substrate for MCT2, the expression of which determines the concentration of NOG that accumulates intracellularly ([NOG]$_{IC}$). In cells with high MCT2 expression, [NOG]$_{IC}$ can reach millimolar levels, which are sufficiently high to additionally engage low–affinity αKG-binding enzymes such as isocitrate dehydrogenase (IDH) and glutamate dehydrogenase (GDH), leading to severely disrupted metabolism and cytotoxicity. Such a polypharmacological mode of action makes it challenging to disentangle the exact mechanisms that account for the effects of NOG on cellular physiology.

In addition to MOG, MCT2 transports endogenous monocarboxylates ranging from pyruvate and lactate to larger ketone bodies such as β-hydroxybutyrate, acetoacetate, α-ketoisovalerate and α-ketoisocaproate[19] (Supplementary Fig. 1a) with a higher affinity than the other SLC16 family members[9]. MCT2 plays important physiological roles including the uptake of astrocyte-secreted lactate into neurons within the brain[20,21]. MCT2 is highly expressed in some human cancers (Supplementary Fig. 1b) and has been proposed as a biomarker for prostate cancer[22], as well as having pro-tumorigenic[23] and pro-metastatic[24] roles in breast cancer. Therefore, there is an emerging need to develop chemical probes to study MCT2 functions.

Here, we report the design and synthesis of MOG-based analogues and use them to explore the MCT2 pharmacophore, and [NOG]$_{IC}$-dependent interference with intracellular targets in the context of their effects on cellular proliferation and survival.

## Results

**Design and synthesis of MOG analogues**. Based on our previous findings[18], we reasoned that MOG could be used as a scaffold to explore both the chemical space accommodated by MCT2 and the cellular roles of αKG-binding proteins. The conversion of DMOG to MOG and subsequently NOG generates compounds with progressively decreased capacity to transverse the plasma membrane (Fig. 1a), so, the stability of MOG and, by extension, that of its analogues, could influence their mode of entry into cells and subsequently the degree of intracellular target engagement. Therefore, with the outlook of generating compounds that could also be used in the future for studies in vivo, we first determined the stability of MOG in whole mouse blood using liquid chromatography-mass spectrometry (LC-MS).

Synthetic MOG was converted to NOG with a half-life that was short and comparable to that of MOG that was transiently generated from DMOG (Fig. 1b, c). Notably, the degradation of DMOG to MOG was even more rapid, with a half-life of just 0.61 minutes. The half-lives of DMOG and MOG in mouse blood are markedly shorter than those previously measured in aqueous solution[18], which we attribute to the well-documented high level of blood esterase activity. These data suggested that increasing stability would be a desirable attribute of MOG analogues.

We designed and synthesised compounds using MOG as the chemical scaffold (**1**, Fig. 1d, e). Based on our previous findings and the fact that esters are typically used as pro-drugs for poorly absorbed carboxylic acid drugs[25–28], we focused on the methyl ester of MOG as the primary cause of compound instability in plasma. Furthermore, as the pharmacophore required for MCT2-driven transport is unknown, substitutions were kept relatively conservative. Therefore, we replaced the glycinate ester on MOG with i) bulkier alkyl esters such as ethyl or isopropyl (compounds **2–3**), ii) esters possessing methyl substituents at the α position (compounds **4–6**), iii) a ketone (compound **7**), or iv) 5-membered aromatic heterocycles (compounds **8–10**). Importantly, compounds **1–6** are predicted to be de-esterified to form NOG or methyl-substituted NOG, and therefore likely also able to engage intracellular targets.

Esters **2** and **3** were designed to minimally increase the steric hindrance of the ester substituent, which would be expected to decrease both chemical and enzyme-mediated instability[29–33]. We also explored branched esters (**4–6**), as substitutions on the α-carbon can increase chemical stability[34] and cellular esterases can exhibit surprising selectivity toward complex esters[35–37]. Ketones (**7**) and amides are classical ester isosteres[38], with the amide typically used to improve the stability of drugs[39]. Notably, such small changes can have a substantial effect on the binding affinity of the compounds to their targets[40]. Finally, 5-membered ring heterocycles, such as oxadiazoles (**10**) or oxazoles (**8**), have also been successfully used as ester bioisosteres[41–44].

**MCT2-dependent uptake by cells is maintained in a subset of MOG analogues**. To determine the MCT2-dependence of MOG analogue uptake into cells, we utilised HCC1569 cells, a human breast cancer line that expresses very low levels of MCT2 and is naturally resistant to MOG-induced toxicity[18]. We compared compound uptake in these cells transduced with a control 'empty vector' plasmid (HCC1569-EV) to an isogenic line that expresses

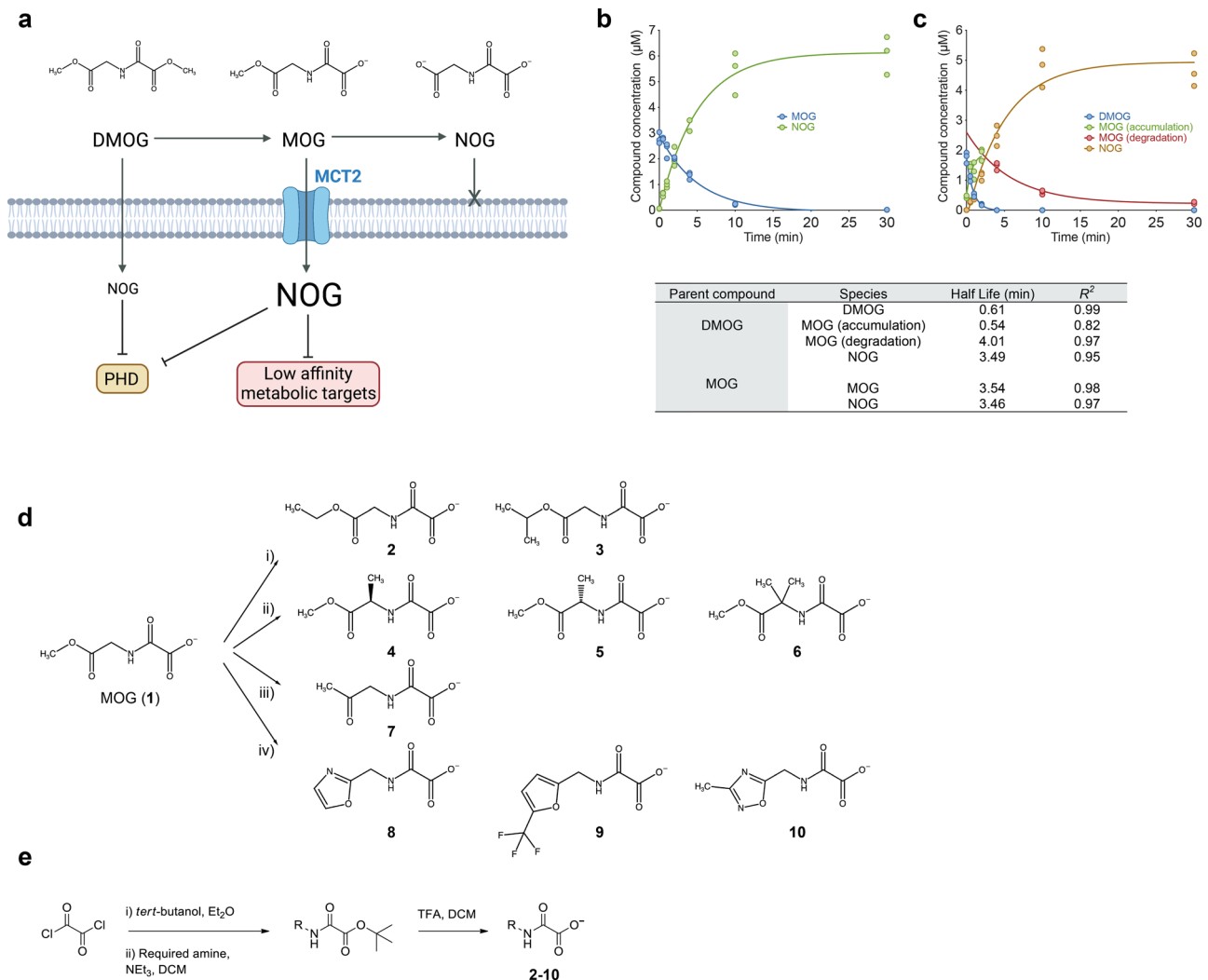

**Fig. 1 Design and synthesis of MOG analogues. a** Schematic (created with Biorender) depicting chemical structures for DMOG, MOG and NOG, their relative cell permeability and cellular targets depending on the intracellular NOG concentrations ([NOG]$_{IC}$) they elicit). DMOG is converted to MOG and subsequently to the active αKG analogue NOG. DMOG is cell-permeable whereas MOG is transported via MCT2 leading to higher [NOG]$_{IC}$ compared to that elicited by DMOG. High [NOG]$_{IC}$ inhibits metabolic enzymes in addition to PHDs. NOG cannot pass through the plasma membrane. **b** Analysis of synthetic MOG stability over time in whole mouse blood by LC-MS ($n = 3$ technical replicates). **c** LC–MS analysis of DMOG stability in whole mouse blood over time ($n = 3$ technical replicates). DMOG is very rapidly converted to MOG, which is also unstable and subsequently forms NOG with similar kinetics to those of synthetic MOG measured in (**b**). Table shows calculated half-lives of DMOG conversion to MOG and subsequently NOG, or of synthetic MOG conversion to NOG from the data shown in panels (**b** and **c**). **d** Structures of MOG glycinate methyl ester replacement analogues designed, synthesised and reported in this work. (i) bulkier alkyl esters (**2**,**3**), (ii) α-methyl substituents (**4–6**), (iii) ketone analogue (**7**), (iv) 5-membered aromatic heterocycles (**8–10**). **e** Synthetic route for the preparation of MOG analogues 2-10 shown in panel (**d**).

exogenous human MCT2 (HCC1569-MCT2) (Fig. 2a, b). After incubation of both cell lines with each analogue for 4 h, we tested whether compounds or derivatives thereof could be detected intracellularly. In the case of compounds **7–10**, we detected the intact parent compound, however, as expected, compounds **1–6** were all de-esterified intracellularly and therefore in these cases we quantified NOG, or the methyl-substituted NOG that was formed. Intact MOG analogues or their products accumulated intracellularly to varying extents, with those derived from the bulkier alkyl esters (**2** and **3**) and α-methyl substituents (**4–6**) reaching concentrations in the millimolar range (Fig. 2c). The de-esterification of compounds **1–6** within cells is expected to further decrease their membrane permeability, which could effectively trap them inside cells and may explain the higher concentrations observed. We defined MCT2-dependent uptake as a two-fold increase in compound accumulation in HCC1569-MCT2 cells

compared with HCC1569-EV cells[45]. The dependence of uptake on MCT2 varied between compound groups but was maintained in both of the bulkier alkyl esters (with the fold-change in uptake of compound **2** being comparable to that of MOG), as well as in all 3 of the 5-membered aromatic heterocycles (compounds **8–10**, Fig. 2d).

Although we found a modest increase in the uptake of the α-methyl and ketone analogues in MCT2-expressing cells compared to controls, these compounds did not meet the two-fold cut off criterion. Since these compounds (**4–7**) harbour very minor modifications of the MOG scaffold, we considered whether they might interact with MCT2 in an inhibitory manner. To test this idea, we assessed the ability of **4–7** to prevent MOG-induced, MCT2-dependent inhibition of respiration[18] in INS1 cells (a rat pancreatic β-cell line with low expression of all endogenous MCT isoforms[46]) that expressed exogenous human MCT2 (INS1-

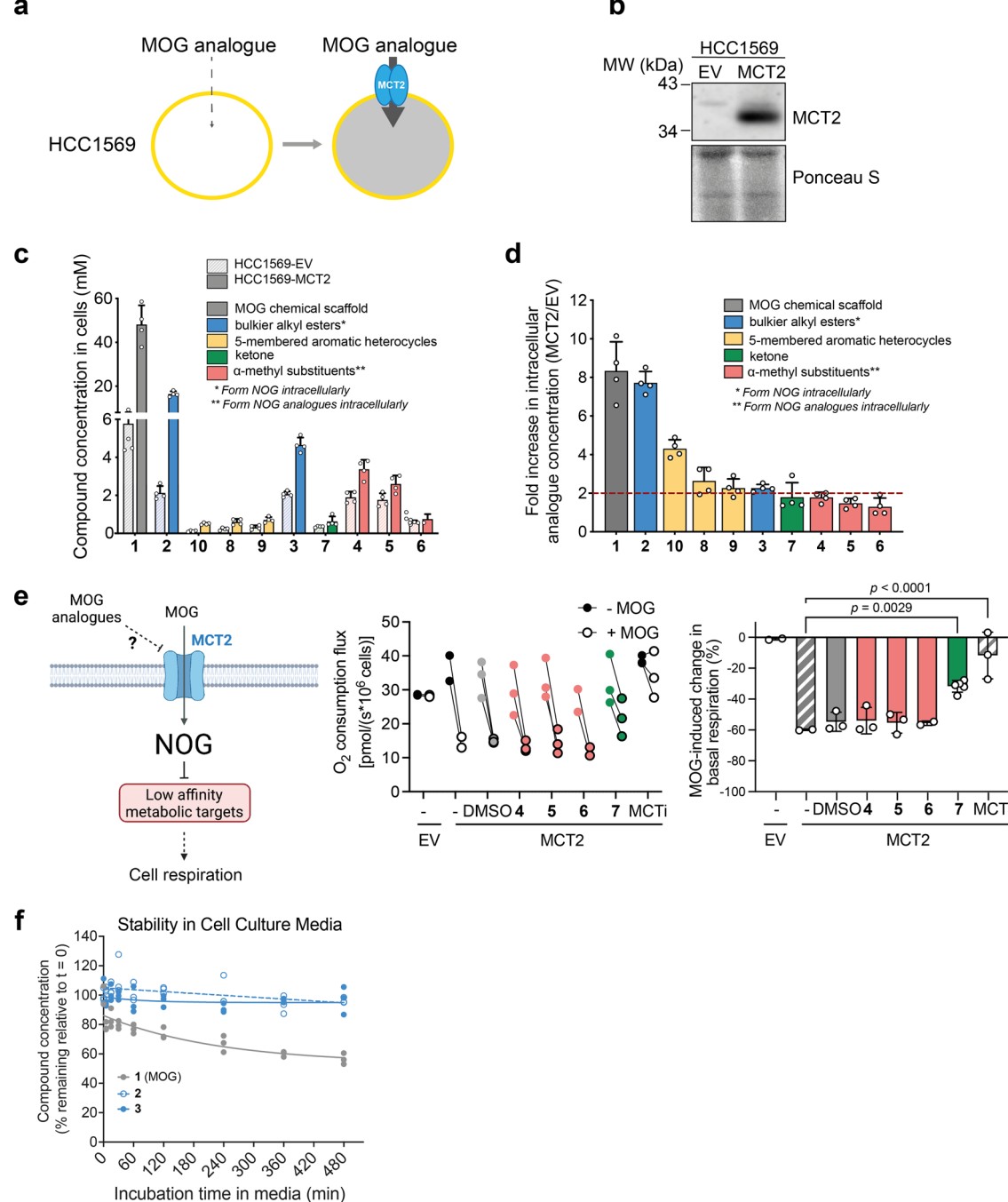

**Fig. 2 MCT2-dependent entry into cells is maintained by alkyl ester and aromatic heterocycle MOG analogues. a** Schematic to illustrate the cell system used to assess dependence of MOG analogue cellular uptake on MCT2. HCC1569 human breast cancer cells were transduced with either an empty pBabePuro vector control (EV) or pBabePuro-MCT2 to stably express exogenous MCT2. **b** Western blot demonstrating expression of exogenous MCT2 in HCC1569-EV or HCC1569-MCT2 cells generated as described in (**a**). **c** Concentration of each of the analogues, or the indicated compounds they produce in cells, in HCC1569-EV or HCC1569-MCT2 cells incubated for 4 h with 1 mM of each of the indicated analogues ($n = 4$ independent wells, mean ± SD). **d** Fold-difference in intracellular concentration of each analogue, or the indicated compounds they produce in cells, in HCC1569-MCT2 cells relative to HCC1569-EV cells. Analogues with a >2-fold (dashed line) increase were considered to be taken up in an MCT2-dependent manner ($n = 4$, mean ± SD). **e** Left: Schematic (created with Biorender) illustrating the strategy for testing analogues **4–7** as putative MCT2 inhibitors). In cells treated with MOG, NOG inhibits metabolic enzymes and thereby leads to decreased respiration. Putative MCT2 inhibitors prevent MOG entry and are expected to attenuate MOG-induced inhibition of respiration. Right: Mean ± SD change in basal cellular respiration (calculated from the data shown in the middle panel) after treatment of INS1-EV or INS1-MCT2 cells with MOG in the presence or absence of the indicated MOG analogues. MOG does not inhibit respiration in the absence of exogenous MCT2 expression illustrating the specificity of the assay. AR-C155858 was used as a positive control for MCT2 inhibition. The ketone analogue **7** attenuates MOG-induced inhibition of respiration consistent with this compound being an MCT2 inhibitor. Significance tested using a one-way ANOVA with Dunnett's test for multiple comparisons ($n = 2–5$ independent measurements). **f** LC-MS analysis to assess stability of MOG or the bulkier alkyl MOG analogues **2** and **3** in cell culture media over time ($n = 3$ independent replicates).

MCT2)(Fig. 2e and Methods). Upon treatment with MOG, basal respiration of INS1-MCT2 (but not EV control) cells decreased by 60%. AR-C155858, a previously described inhibitor of MCT2[47] almost completely prevented inhibition of respiration by MOG. Addition of the α-methyl substituents had no effect on MOG-induced decrease of respiration, however, co-incubation with compound **7** decreased the inhibitory effect of MOG by half. Importantly, none of the analogues alone inhibited respiration (Fig. 2e, middle panel). This finding suggests that the replacement of the glycinate ester with a ketone group converts MOG from a substrate to an inhibitor of MCT2.

To test whether the difference in $[NOG]_{IC}$ in cells treated with analogues **2** and **3** can be accounted for by altered stability and thereby compound availability to cells, we measured the conversion of these compounds to NOG in cell culture media. Both compounds demonstrated similar, improved stability compared to MOG (Fig. 2f), suggesting that increased $[NOG]_{IC}$ in cells treated with compound **2** compared to compound **3** is likely due to differences in transport or intracellular de-esterification rates rather than differences in their stability in media. Interestingly, the half-life of compound **3** in blood was significantly longer than that of either MOG or compound **2**, also mirrored by the greater persistence of **3** compared to MOG in the blood of animals administered with these compounds (Supplementary Fig. 2b).

In summary, we generated MOG analogues with differential dependence on MCT2 for cell entry and with the ability to generate a range of $[NOG]_{IC}$. Further investigations were focused on compounds that showed MCT2-dependent uptake.

**MCT2-dependent cellular effects of MOG analogues are proportional to the $[NOG]_{IC}$ they elicit.** MOG inhibits cell proliferation and leads to apoptosis in MCT2-expressing cells in a manner that depends on $[NOG]_{IC}$[18]. We therefore assessed whether the bulkier alkyl esters and 5-membered aromatic heterocycles maintain the ability to elicit cytotoxicity in our HCC1569 ± MCT2 cell model. Over 96 h, MOG inhibited cell mass accumulation in a concentration-dependent manner (Fig. 3a). This inhibition was markedly higher in MCT2-expressing cells and was associated with increased apoptosis. In contrast, although the bulkier alkyl esters (**2**, **3**) also slowed cell mass accumulation to a larger extent in HCC1569-MCT2 than HCC1569-EV cells, they did not elicit notable apoptosis except at the highest concentration. The cytostatic effects of analogues **2** and **3** in HCC1569-MCT2 cells were proportional to the observed $[NOG]_{IC}$ achieved after treatment with these compounds, respectively (Fig. 3b). The 5-membered aromatic heterocycles (**8–10**) did not affect cell mass accumulation, either in the presence or absence of MCT2, except compound **9**, which, at the highest dose, led to a small increase in apoptosis.

We also tested cellular effects and uptake of our analogues in LN229 (glioblastoma), MCF7 (breast cancer) and SN12C (kidney cancer) cells that express endogenous MCT2[18]. MOG attenuated cell mass accumulation in all three cell lines (Fig. 3c). Compounds **2** and **3** caused either a small or no decrease in cell mass accumulation and led to much lower $[NOG]_{ic}$ relative to that achieved with MOG (Fig. 3d). Notably, **2** and **3** elicited $[NOG]_{ic}$ in these cells that was almost 10-fold lower than the $[NOG]_{ic}$ elicited in HCC1569-MCT2 cells (Fig. 3b), likely explaining the attenuated cytostatic effects of these compounds in cells with endogenous MCT2 expression levels relative to MCT2-overexpressing cells.

Together, these findings demonstrated that substitution of the methyl-ester of MOG with bulkier alkyl esters created compounds that, under equivalent treatment conditions, yield lower

intracellular NOG levels and lower or no cytotoxicity compared to MOG.

**Bulkier alkyl-ester MOG analogues have attenuated effects on metabolism compared to MOG.** MCT2 expression promotes MOG-induced cytotoxicity by eliciting metabolic changes due to high $[NOG]_{ic}$[18]. We hypothesised that the lack of cytotoxic effects by MCT2-dependent MOG analogues (**2**, **3**, **8–10**) was linked to a decreased ability to perturb metabolism. To test this idea, we treated cells for 4 h with MOG analogues and analysed their metabolism by gas chromatography-mass spectrometry (GC–MS). As previously described[18], MOG caused a characteristic MCT2-dependent decrease in TCA cycle intermediates and increase in amino acid concentrations (Fig. 4a and Supplementary Fig. 3a, b). Consistent with our hypothesis, this metabolic signature was notably dampened in cells treated with any of the analogues tested (Supplementary Fig. 3a, b). Similarly, cells with endogenous levels of MCT2 (MCF7, SN12C and LN229) showed an attenuated metabolic response to analogues **2** and **3** compared to MOG (Fig. 4a).

Metabolic changes induced by high $[NOG]_{IC}$ are, in part, driven by inhibition of GDH and IDH. In cells labelled with $[U^{13}C]$-glutamine, the extent of GDH or IDH inhibition can be determined by monitoring, respectively, levels of the citrate m + 4 isotopologue formed from the oxidative use of glutamine carbons in the canonical TCA cycle and the citrate m + 5 isotopologue produced by the reductive carboxylation of αKG (Fig. 4b)[18].

Treatment of cells with **2** or **3** did not affect citrate m + 4 labelling and caused a modest MCT2-dependent decrease in citrate m + 5, which was, however, significantly less pronounced than that caused by MOG (Supplementary Fig. 3c). The analogues demonstrated similarly attenuated effects on $[U^{13}C]$-glutamine-derived citrate labelling in SN12C, MCF7 and LN229 cells (Fig. 4c). Notably, the modest decrease in citrate m + 5 was more pronounced with **2** than with **3**, reflecting the higher $[NOG]_{IC}$ in cells treated with the former (Fig. 3d).

Together, these metabolic analyses support the idea that lower $[NOG]_{IC}$ elicited by bulkier alkyl-MOG analogues does not suffice to fully inhibit glutamine metabolism, and, together with previous observations[18], explain their decreased ability to induce cytotoxicity.

**MOG analogues retain ability to inhibit PHDs and help uncouple their activity from regulation of mTORC1 by glutaminolysis.** DMOG inhibits PHDs and thereby promotes stabilisation of HIF1α at lower $[NOG]_{IC}$ than those required to inhibit glutaminolysis due to the higher affinity of NOG for PHDs than for metabolic targets[18]. To test whether the low $[NOG]_{IC}$ we found with MOG analogues suffice to stabilise HIF1α, we treated cells for 4 h with each analogue at 1 mM (a typical concentration at which DMOG is used to stabilise HIF1α in cell culture studies) and monitored HIF1α levels by western blot. In HCC-1569 ± MCT2 cells, either of the alkyl esters (**2**, **3**), which produce NOG intracellularly, induced HIF1α stabilisation (Fig. 4d) in an MCT2- and $[NOG]_{IC}$-dependent manner. Conversely, the aromatic heterocycles (**8–10**) did not induce HIF1α stabilisation, suggesting that despite the conservation of the oxoacetate moiety of NOG, the addition of an aromatic group on the glycinate site is incompatible with inhibition of PHDs at the compound concentrations we used. In MCF7 cells, compounds **2** or **3** stabilised HIF1α with kinetics similar to those of MOG (Fig. 4e). Importantly, the protein levels of lactate dehydrogenase A (LDHA) and pyruvate kinase M2 (PKM2), two prototypical HIF1α gene targets, were equally upregulated in response to treatment with compounds **2** and **3**, and with comparable kinetics. These data

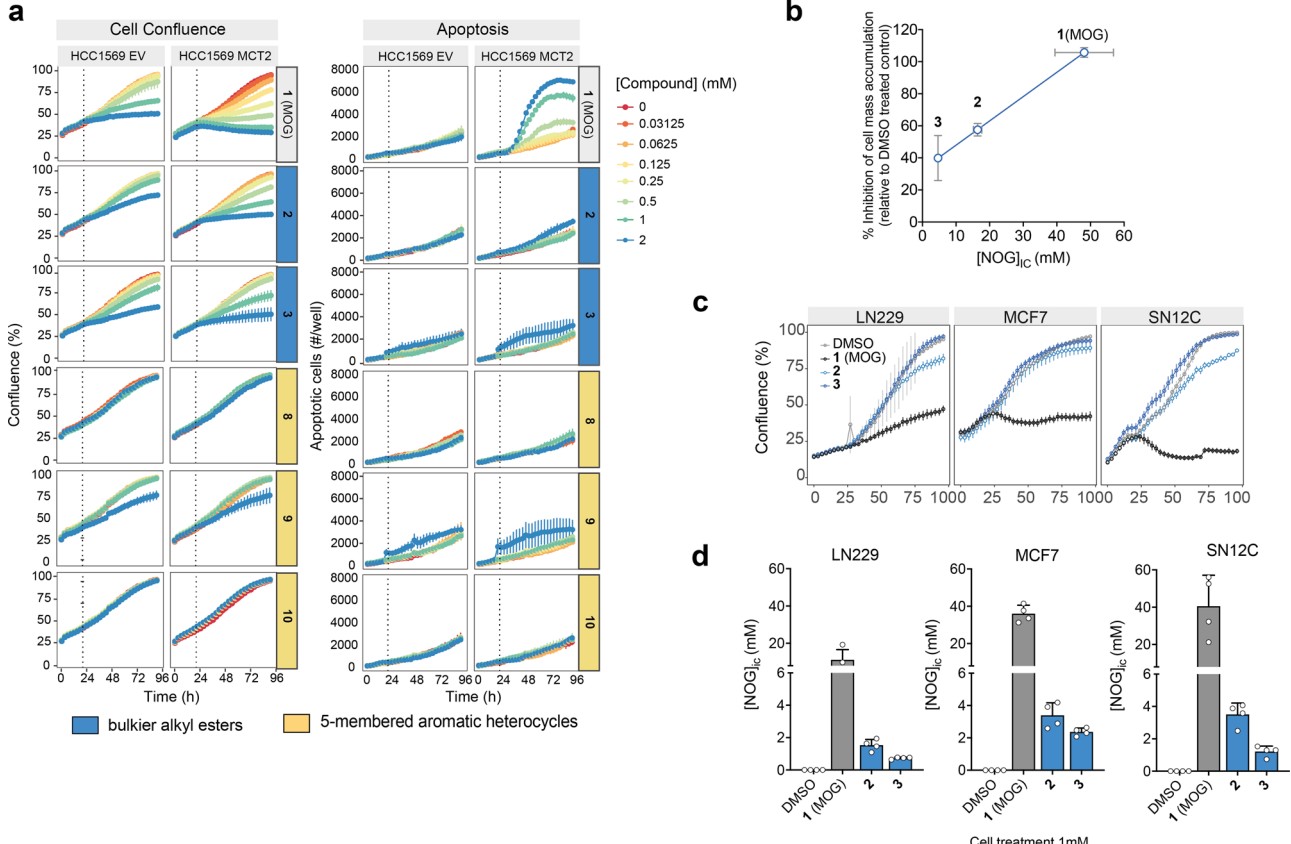

**Fig. 3 Analogues elicit lower [NOG]$_{IC}$ and decreased cytotoxicity compared to MOG. a** Confluence and apoptosis measurements, over time, of HCC1569-EV or HCC1569-MCT2 cells in the presence of MOG analogues added to cells at the indicated concentrations at 20 h (dotted line) ($n = 3$ independent wells, mean ± SD). **b** Degree of inhibition of cell mass accumulation after treatment with 1 mM of the indicated analogues (data from panel **a**) is proportional to the corresponding [NOG]$_{IC}$ elicited by each analogue. Error bars represent ± SD. **c** Confluence, over time, of MCF7, SN12C or LN229 cells in the presence of the indicated compounds added to cells at 16 h ($n = 3$ independent wells, mean ± SD). **d** [NOG]$_{IC}$ in MCF7, SN12C or LN229 cells treated with 1 mM of each of the indicated for 4 h ($n = 4$ independent wells, mean ± SD).

showed that, even though analogues **2** and **3** lead to lower [NOG]$_{IC}$ than MOG, they stabilise HIF1α to the same extent as MOG in cells with endogenous expression of MCT2.

In addition to regulation of HIF1α, PHDs have also been reported to mediate glutaminolysis-fuelled mTORC1 activation[48]. Given that MOG analogues fail to inhibit glutaminolysis but can still inhibit PHDs, we compared their effects on ribosomal protein S6 kinase (S6K) phosphorylation (a typical readout of mTORC1 activity) to those of MOG, which can inhibit both glutaminolysis and PHDs. Even though **2** and **3** could inhibit PHDs (as shown by HIF1α stabilisation) they failed to inhibit mTORC1 signalling after 4 or 8 h of treatment (Fig. 4e). All three compounds suppressed S6K phosphorylation after 24 h suggesting this latent mTORC1 inhibition is likely secondary to HIF1α activation rather than a direct effect of the analogues. Therefore, comparison of the effects of analogues to MOG revealed that the inhibitory effects of (D)MOG on mTORC1 signalling are likely due to attenuated glutaminolysis rather than inhibition of PHDs. Consistent with this idea, the more specific PHD inhibitor FG4592 failed to suppress mTORC1 activation after re-feeding starved cells with amino acids (a prototypical mTORC1 activator) (Supplementary Fig. 3d).

In summary, our data showed that compounds **2** and **3** led to inhibition of PHDs but caused minimal metabolic effects, cytotoxicity and mTORC1 inhibition compared to MOG, thus enabling us to uncouple the cellular effects of NOG elicited by its metabolic targets from those that occur due to PHDs.

## Discussion

Metabolism has far-reaching effects on cellular physiology that extend beyond biomass accumulation, energy production and redox balance. A prototypical example of this concept is α-KG, a central metabolic node that is not only the entry point of glutamine-derived carbons into the TCA cycle, but also has important regulatory roles for key signalling proteins such as mTOR and HIF1α[8,15]. Furthermore, αKG acts as a cofactor for DNA- and chromatin-modifying enzymes such as TET hydroxylases[9] and Jumonji demethylases[7]; consequently, fluctuations in the concentration of αKG can also influence epigenetic processes, leading to long lasting effects within the cell. NOG is a structural analogue of αKG that has been used to help understand many of the established roles of this important metabolite. DMOG is a membrane-permeable NOG ester that is rapidly de-esterified in cell culture media to the monocarboxylate MOG, a substrate of the transporter MCT2. The expression level of MCT2 determines the [NOG]$_{IC}$, which, at high levels, inhibits a number of low affinity metabolic targets such as GDH and IDH, leading to toxicity in MCT2-expressing cancer cells[18].

In addition to its in vitro use, DMOG has been extensively used, primarily as a pharmacological stabiliser of HIF1α, in vivo for pre-clinical studies[49,50] where, typically, it is administered at concentrations that far exceed those required to inhibit the intended intracellular targets[11,51,52]. DMOG instability as a result of chemical or enzymatic de-esterification and a subsequent loss of cell-permeability could explain the disparity in potency observed

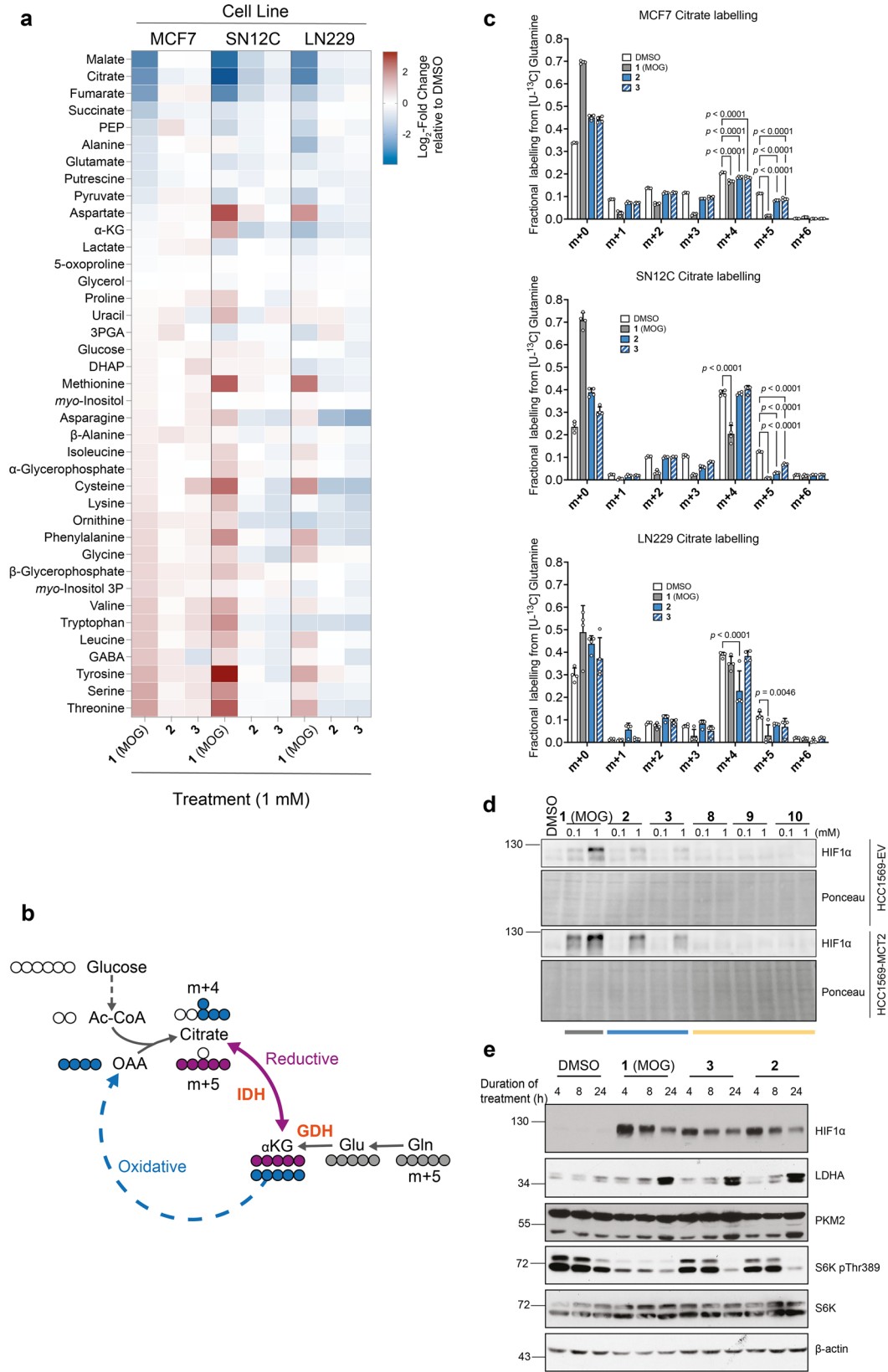

between in vitro (purified enzyme) and in vivo studies, particularly in light of the high level of esterase activity in blood. In support of this hypothesis, here we show that DMOG and MOG are both rapidly converted to NOG in blood, each with a half-life of less than 5 minutes. This poor stability in blood should therefore be a key consideration when using DMOG as a tool compound in vivo.

MCT2 has a number of established physiological as well as pathological roles yet is one of the lesser-studied members of the monocarboxylate transporter family. A more detailed mechanistic understanding of this transporter could therefore open up new therapeutic opportunities and provide the basis for further studies to generate in vivo imaging tools. The new MOG analogues we

**Fig. 4 MOG analogues help deconvolute cellular effects of NOG elicited by inhibition of metabolic targets from those due to inhibition of PHDs. a** Heat map showing $\log_2$ fold-changes in the mean abundance of the indicated metabolites in MCF7, LN229 and SN12C cells treated for 4 h with the indicated compounds relative to DMSO (vehicle)-treated controls. Metabolites are ordered from the highest to the lowest fold-change values in the MCF7 MOG-treated condition. **b**. Schematic to demonstrate different routes of citrate synthesis and subsequent labelling patterns from [U-$^{13}$C]-Glutamine. **c** Labelling of citrate from [U-$^{13}$C]-Glutamine in MCF7, LN229 or SN12C cells treated with 1 mM of each of the indicated analogues for 4 h. $n = 4$ independent wells, mean ± SD; significance tested by multiple t-tests with Holm-Sidak correction for multiple comparisons. (* = $p < 0.05$, ** = $p < 0.01$, *** = $p < 0.001$). **d** Western blot showing HIF1α protein expression in HCC1569-EV or HCC1569-MCT2 cells treated with 0.1 or 1.0 mM of the indicated analogues, or with DMSO for 4 h. **e** Western blot showing protein levels of HIF1α, the HIF1α target gene protein products LDHA and PKM2, and the mTORC1 kinase substrate S6K (total and phosphorylated at Thr389) in lysates of MCF7 cells treated with 1 mM of the indicated compounds for 4, 8 or 24 h.

report here helped us to further explore the structure-activity relationship (SAR) between MCT2 and its ligands, beyond what has been established based on endogenous substrates[19].

Interestingly, we observed very little tolerance for the α-methyl substitutions (**4**–**6**), all of which failed to meet our 2-fold threshold for MCT2-dependent uptake. These three analogues bare some structural similarity to α-ketoisocaproate (Fig. 1d, Supplementary Fig. 1a), for which MCT2 has a $K_m$ of 100 µM[19]. Their lack of transport therefore suggests that the combination of an α-methyl substitution with a carboxyl-ester group cannot be accommodated by MCT2 (Fig. 2c, d). Unexpectedly, while we also observed no MCT2-dependent transport of the ketone analogue (**7**), this compound prevented a MOG-induced decrease in cellular respiration, suggesting it can inhibit MCT2 transport activity (Fig. 2e). This finding could potentially indicate that though **7** can still bind to MCT2, the oxygen within the MOG ester participates in an interaction within the substrate binding pocket that is required for transport.

MCT2-dependent transport was maintained in the bulkier alkyl esters. We found that replacement of the methyl-ester leaving group with an ethyl-ester (**2**) was well-tolerated by MCT2, with an almost eight-fold increase in uptake by MCT2-expressing cells compared to the control cell line. MCT2-dependent transport was maintained with an isopropyl-ester substitution (**3**), however, it was lower compared to **2** indicating that the increased size of the substitution led to some steric hindrance within the transporter.

The 5-membered aromatic heterocycles (**8**–**10**) were also all transported in an MCT2-dependent manner with between a 2- and 4-fold enrichment in MCT2-expressing cells. Given the increasing interest in the role of MCT2 in cancer[22–24], this finding provides a useful set of scaffolds for the development of ligands to image MCT2 activity in vivo.

Even in the case of very selective drugs, intracellular concentrations higher than those required to engage the intended target could lead to off-target effects and toxicity. Here, we demonstrate that transporter-mediated uptake determines intracellular concentration of compounds and thereby dictates their efficacy and toxicity. Although both the bulkier alkyl esters (**2**, **3**) are converted to NOG intracellularly, the [NOG]$_{IC}$ achieved in MCT2-expressing cells varied widely (48.1, 16.4 and 4.65 mM for MOG, **2** and **3**, respectively), despite each analogue being dosed at the same concentration. The [NOG]$_{IC}$ achieved by each compound determined their effects within cells, as reflected by their relative impact on cell mass accumulation and apoptosis (Fig. 3a, b). PHD engagement by **2** and **3** was maintained across a range of cell lines with both over-expressed and endogenous levels of MCT2, based on the observed stabilisation of HIF1α (Fig. 4d, e). Similarly, compounds **2** and **3** only partially inhibited reductive carboxylation and did not significantly suppress the oxidative production of citrate (Fig. 4c, Supplementary Fig. 3c) via inhibition of GDH[18]; as such, these analogues only minimally depleted TCA intermediates relative to the depletion seen when dosing with MOG (Fig. 4a, Supplementary Fig. 3a). Together, our observations suggest that the lack of cytotoxicity in cells treated with **2** and **3** is because these compounds result in a [NOG]$_{IC}$ that is not sufficient to engage all the NOG targets that are collectively required for the cellular effects seen with MOG.

Given the extensive roles of αKG in cells, it is widely appreciated that NOG, as an αKG mimic, is a promiscuous compound. Significant efforts have been made to generate tool compounds and potential clinical leads to inhibit a number of its targets, and in particular the prolyl hydroxylases[11], more selectively. The differential transport of our analogues and subsequent differences in [NOG]$_{IC}$ have enabled us to better understand the mechanism of action of (D)MOG. Previous studies implicated PHDs in the regulation of the mTORC1 pathway, in part by demonstrating that DMOG inhibits mTORC1 activity[48]. However, since glutaminolysis is also known to activate mTORC1[15], the simultaneous actions of (D)MOG on both glutaminolysis and PHD activity complicate these conclusions. We demonstrate here that while treatment of cells with MOG leads to rapid inhibition of mTORC1 signalling, compounds **2** and **3**, which inhibit PHD activity but do not recapitulate the metabolic effects of MOG, are unable to inhibit mTORC1 signalling. Notably, in muscle, PHD1 has been shown to regulate mTORC1 independently of its enzymatic activity[53]. We cannot exclude the possibility that PHDs regulate mTORC1 in other physiological contexts, however, our findings suggest that, under the experimental conditions we used, inhibition of PHD enzymatic activity, alone, is insufficient to impact mTORC1 activity.

For αKG-dependent dioxygenases beyond PHDs, there are far fewer specific chemical inhibitors available. The TET enzymes are of particular interest, given their well-established roles in regulating DNA methylation during early embryonic development. More recently, it has become clear that these enzymes also mediate the effects of cellular metabolic state upon epigenetic regulation[54,55], which, in turn, can influence differentiation in cancer[16]. Isoform-specific TET inhibitors have yet to be developed, and so 'bump and hole'-based approaches[56] have been employed to allow individual isoform targeting via engineered enzyme isoforms with expanded active sites that can accommodate bulkier NOG analogues[57]. We speculate that our work could aid in the creation of cell-permeable derivatives of these NOG analogues, enabling the study of specific TET enzymes both in cells and potentially also in vivo.

Finally, to enable the study of MCT2-specific interactions our analogues were designed to mimic MOG. However, our findings could also be of use, more generally, for the many labs that use DMOG as a tool. Further development of compound **3** with analogues that also feature bulkier oxoacetate carboxyl ester groups will likely enhance blood stability while maintaining general membrane permeability, thereby further improving the pharmacokinetic profile of this tool compound.

## Methods

**Chemical Synthesis**. Please see Supplementary Methods.

**Cell lines and cell culture**. HCC1569, MCF7, LN229 and SN12C cells were obtained from the American Type Culture Collection. Cells were cultured in RPMI

1640 medium (Gibco, 31840) containing 10% fetal calf serum (FCS), 2 mM glutamine and 100 U/ml penicillin/streptomycin, and were incubated in a humidified incubator at 37 °C and 5% $CO_2$. All cell lines tested mycoplasma-free, and identity was confirmed by short-tandem-repeat profiling (Francis Crick Institute Cell Services Science Technology Platform). Generation of HCC1569-MCT2 overexpressing cells was achieved using retroviral transduction of HCC1569 cells with a pBabe-puro vector containing the *SLC16A7* cDNA sequence, as described previously[18]. HCC1569-EV cells transduced with an empty pBabe-puro vector were used as controls.

**Western blotting**. Cells on cell culture dishes were washed twice with PBS, before scraping in SDS sample buffer (without beta-mercaptoethanol or bromophenol blue) and boiled for 5 min at 95 °C. Protein was quantified using a BCA assay before adding beta-mercaptoethanol and bromophenol blue and resolving by SDS–PAGE. Proteins were transferred to nitrocellulose membranes by electroblotting, before blocking with 5% milk in Tris-buffered saline (50 mM Tris-HCl, pH 7.5, and 150 mM NaCl) containing 0.05% Tween 20 (TBS-T). Membranes were then incubated with the primary antibody overnight at 4 °C, washed with TBS-T and incubated with horseradish peroxidase-conjugated secondary antibody for 1 h at RT in 5% milk TBS-T. Antibodies were visualized by chemiluminescence and imaged using an Amersham Imagequant 600 RGB.

Primary antibodies used were obtained from Cell Signalling Technology: P-S6 kinase #9234 1:1000, S6 kinase #2708 1:2000, LDHA: #2012 1:1000, PKM2 #3198 1:1000; Sigma: β-actin A5316 1:1000; BD Biosciences: HIF1α #610959 1:250; MCT2 1:500 (generated by the Anastasiou lab). Secondary antibodies were goat anti-rabbit or anti-mouse IgG from Millipore (#AP132P, #AP127P respectively).

Uncropped versions of all the western blots used in this study can be found in Supplementary Fig. 4.

**Cell confluence and apoptosis measurements**. Cell proliferation and apoptosis were measured in real time using an IncuCyteZoom (Essen Bioscience). Cell lines were seeded in 96-well plates at between 4,000 and 9,000 cells per well (depending on growth rate), in the presence of Incucyte Caspase 3/7 Green Apoptosis Assay Reagent (Essen Bioscience, used according to manufacturer's instructions). MOG analogues were added at the indicated doses 16–20 h after seeding. The IncuCyteZoom was programmed to image cells (phase and fluorescence) at 3 h intervals, and automated image analysis was used to determine confluence and number of apoptotic cells.

**Assessing ability of analogues to inhibit MCT2-mediated MOG-induced cellular respiration**. Oxygen consumption was measured in intact INS1 cells that stably expressed human MCT2 or an empty vector control using an Oroboros Oxygraph-2K oxygen electrode system (Oroboros Instruments) at 37 °C. Cells from one confluent 10 cm cell culture dish were used per replicate, per condition. After trypsinisation, cells were resuspended in Hank's buffered saline solution (HBSS) and incubated with 0.1% DMSO or 1 mM of the indicated analogue for 30 mins before the start of oximetry. Under each treatment condition, following an initial measurement of basal oxygen consumption, 0.25 mM MOG were added to the cell suspension in the oximeter chamber. Oxygen consumption was normalised for cell number. Inhibition of MCT2 was determined by the ability of analogues to prevent a MOG-induced decrease in cellular respiration.

**Stable isotope labelling and metabolite extraction for metabolomics**. Cells were seeded 1 day prior to the experiment in 6-cm dishes in RPMI medium (as described above), containing dialysed FCS (3,500-Da MWCO). Medium was replaced with fresh at $t = -1$ h. At $t = 0$, medium was replaced again to medium containing [U-$^{13}$C]-glutamine (2 mM) and the MOG analogue of interest (1 mM) or 0.1% DMSO (vehicle control). Treatment with compounds and labelling was carried out for 4 h unless otherwise stated. Four or five technical-replicate plates were used per condition and two or three additional plates of each cell line were counted to use for normalisation of metabolite measurements. Cell diameter was also recorded for calculation of cell volumes in order to determine intracellular concentrations. Cell diameter and number were measured using a Nexcelcom Bioscience Cellometer Auto T4. At the end of the experiment, plates were washed twice with ice-cold PBS, before quenching cells with the addition of 725 μl dry-ice-cold methanol. Each plate was then scraped on ice, and samples were transferred to a microcentrifuge tube containing 160 μl $CHCl_3$ and 180 μl $H_2O$ (containing 2 nmol of scyllo-inositol as an internal standard). Plates were scraped once more with an additional 725 μl of cold MeOH, which was then added to the microcentrifuge tube containing the rest of the sample. Samples were sonicated for $3 \times 8$ min in a water bath, and metabolites were extracted overnight at 4 °C. Precipitated material was removed by centrifugation and samples were subsequently dried and resuspended in 3:3:1 (vol/vol/vol) MeOH/$H_2O$/$CHCl_3$ (350 μl total), to separate polar and apolar metabolites into an upper aqueous phase and lower organic phase, respectively.

**Analogue uptake assays**. Cells were incubated with MOG or MOG analogues (1 mM) for 4 h. Cells were then washed with ice-cold PBS, and extracted as described for GC–MS above. Samples of the polar phases were diluted 50-fold in

1:1 (vol/vol) MeOH/$H_2O$ (containing 5 μM [U-$^{13}$C,$^{15}$N]-valine as an internal standard) and analysed by LC–MS as described below.

**Gas Chromatography-Mass Spectrometry**. For GC–MS analysis, 150 μl of the aqueous phase was dried down in a vial insert, before washing twice with 40 μl MeOH and drying again. Samples were methoximated (20 μl of 20 mg/ml methoxyamine in pyridine, RT overnight) before derivatising with 20 μl of N,O-bis(trimetylsilyl)trifluoroacetamide + 1% trimethylchlorosilane (Sigma, 33148) for ≥1 h. An Agilent 7890B-5977A GC–MS system was use to perform metabolite analysis. Splitless injection (injection temperature 270 °C) onto a 30 m + 10 m × 0.25 mm DB-5MS + DG column (Agilent J&W) was used, using helium carrier gas, in electron-impact ionization (EI) mode. Initial oven temperature was 70 °C (2 min) with a subsequent increase to 295 °C at 12.5 °C/min, then to 320 °C at 25 °C/min (before holding for 3 min). Metabolite identification and quantification was performed using MassHunter Workstation software (B.06.00 SP01, Agilent Technologies) or MANIC software, an in-house developed adaptation of the GAVIN package[58], by comparison to the retention times, mass spectra and responses of known amounts of authentic standards. Fractional labelling of individual metabolites is reported after correction for natural abundance.

**Liquid Chromatography-Mass Spectrometry**. The LC–MS method was adapted from ref. [59]. Samples were injected into a Dionex UltiMate LC system (Thermo Scientific) using a ZIC-pHILIC (150 mm × 4.6 mm, 5-μm particle) column (Merck Sequant). A 15-min elution gradient was used (80% solvent A to 20% solvent B), followed by a 5-min wash (95:5 solvent A to solvent B) and 5-min re-equilibration; solvent A was 20 mM ammonium carbonate in water (Optima HPLC grade, Sigma Aldrich) and solvent B was acetonitrile (Optima HPLC grade, Sigma Aldrich). Flow rate, 300 μl/min; column temperature, 25 °C; injection volume, 10 μl; and auto-sampler temperature, 4 °C. MS was performed with positive/negative polarity switching using a Q Exactive Orbitrap (Thermo Scientific) with a HESI II (heated electrospray ionization) probe. MS parameters: spray voltage, 3.5 kV and 3.2 kV for positive and negative modes, respectively; probe temperature, 320 °C; sheath and auxiliary gases, 30 and 5 arbitrary units, respectively; full scan range, 70 to 1,050 m/z with settings of AGC target and resolution as 'balanced' and 'high' ($3 \times 106$ and 70,000), respectively. Xcalibur 3.0.63 software (Thermo Scientific) was used to record data. Prior to analysis, mass calibration was performed for both ESI polarities using the standard Thermo Scientific Calmix solution. Calibration stability was enhanced by application of lock-mass correction to each analytical run using ubiquitous low-mass contaminants. Parallel reaction monitoring acquisition parameters: resolution, 17,500; auto gain control target, $2 \times 105$; maximum isolation time, 100 ms; isolation window, m/z 0.4; and collision energies, set individually in high-energy collisional-dissociation mode. Equal volumes of each sample were pooled to provide quality-control samples and were analysed throughout the run, thereby providing a measurement of the stability and performance of the system. Xcalibur Qual Browser and Tracefinder 4.1 software (Thermo Scientific) were used to perform qualitative and quantitative analysis respectively, according to the manufacturer's workflows.

**Blood stability assay**. Blood was collected from euthanised 10–12 week old NSG female mice into heparinised tubes and used immediately for experiments. To test stability, compounds were incubated at a final concentration of 100 μM in blood (600 μL total volume), pre-warmed to 37 °C Samples were collected in triplicate at 0 (compound added into sample after extraction), 5, 15, 30 and 60 minutes. At the indicated time, 30 μl was taken and added to a tube on ice containing 100 μl chloroform, 300 μl MeOH and 270 μl $H_2O$. [U$^{13}$C-$^{15}$N] valine was present in the aqueous phase at a final concentration of 5 μM. Immediately after collection, samples were vortexed and placed on ice. After all samples had been collected, they were sonicated for $3 \times 8$ mins and incubated for 1 h at 4 °C to allow extraction to proceed. Samples were then centrifuged (10 min, 4 °C, full speed) and the aqueous phase transferred to a new tube, and stored at -80 until they were ready to run on the LC-MS (Q Exactive) system, as described above. Samples were quantitated against a 7-point standard curve of compound in mouse blood extract to minimise matrix effects.

**Assessment of compound stability in the blood of mice in vivo**. All experimental procedures using mice were conducted in conformity with public health service policy on humane care and use of laboratory animals, in accordance with Animals (Scientific Procedures) Act 1986 and the GSK Policy on the Care, Welfare and Treatment of Animals, approved by The Francis Crick Institute's Animal Welfare and Ethical Review Body (AWERB) and comply with a license to the Anastasiou laboratory ratified by the UK Home Office. Adult (10–12 week old) female NSG mice were dosed by intra-peritoneal injection with 1 (MOG) or compound **3** at 100 mg/kg. Blood samples were collected from the saphenous vein onto a Mitra micro-sampling device (10 μl capacity, 100601-A, neoteryx) prior to injection ($t = 0$) and at 15 min, 1 h and 2 h post injection. At the end of the experiment, animals were euthanised by cervical dislocation and tissues were harvested immediately and snap-frozen in liquid nitrogen before storing at -80 °C. For blood samples, the micro-sampler 'sponge' tip was transferred into a microfuge tube containing 33.3 μl $CHCl_3$, 100 μl MeOH and vortexed thoroughly. [U$^{13}$C-$^{15}$N]-valine was included to a final concentration of 5 μM in aqueous phase. Samples were

sonicated 3 × 8 mins at 4 °C in sonicating water bath and stored at -80 until required. Phases were split by addition of 90 μl $H_2O$ before spinning at 4 °C full speed for 5 mins (since MOG analogues are susceptible to aqueous hydrolysis, $H_2O$ was added immediately prior to running). Aqueous phase was run on LC–MS as described above.

**Statistics and reproducibility.** Replicate type and numbers are stated within each figure legend. Statistical analyses throughout this work were performed using GraphPad Prism® 7.0b or later versions. Comparisons were made using either unpaired t-tests, multiple t-tests with the Holm-Sidak method for multiple comparison testing, one-way ANOVA with Dunnett's correction for multiple comparisons or two-way ANOVA with Tukey's test for multiple comparisons, as indicated in the respective figure legends.

**Reporting summary.** Further information on research design is available in the Nature Research Reporting Summary linked to this article.

## Data availability

The results shown in Supplementary Fig. 1b are based upon data generated by the TCGA Research Network (https://www.cancer.gov/tcga). The source data underlying graphs shown in this study are provided in Supplementary Data 1. Uncropped versions of the western blots used in this study (Figs. 2b, 4d, 4e, and Supplementary Fig. 3d) can be found in Supplementary Fig. 4.

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

## Acknowledgements

We thank all members of the D.A. laboratory for valuable discussions and input throughout this work, members of the laboratory of L.F. for critical reading of the manuscript and the Crick Translation team for useful discussions. L.F.'s laboratory is funded by the MRC (MC-A654-5QC70). This work was funded by the MRC (MC_UP_1202/1) and by the Francis Crick Institute which receives its core funding from Cancer Research UK (FC001033), the UK Medical Research Council (FC001033) and the Wellcome Trust (FC001033) to D.A. For the purpose of Open Access, the authors have applied a CC BY public copyright licence to any Author Accepted Manuscript version arising from this submission.

## Author contributions

S.C. and D.H. designed (with input from L.F. and D.A.) and synthesised MOG analogues, and advised on mouse dosing experimental design; N.B. assisted with cell line work and related western blots and performed cellular respiration experiments together with P.N.; P.N. also assisted with compound dosing in mice; E.S. assisted with and advised on compound stability measurements; M.S.d.S. and J.I.M. assisted with and advised on metabolomics experiments; L.F. designed and performed all other experiments, analysed and interpreted data. D.A. supervised the study, designed experiments and interpreted data. L.F. wrote the first draft of the manuscript and developed it with support from D.A. and input from S.C. and D.H. All authors reviewed and commented on the manuscript.

## Funding

## Competing interests

The authors declare no competing interests.
