## [Peer Review File · Communications Biology]

Reviewers' comments:

Reviewer #1 (Remarks to the Author):

Fets et al have demonstrated the effect of new MOG analogues in terms of stability and MCT2 biology.

However, some suggestions to improve this manuscript are:

1. Please improve the quality of Figure 3c.
2. The source (company information) of secondary antibodies should be included.
3. Cell proliferation and apoptosis measurements (in materials and methods section) requires substantial elaboration.
4. Figure 4B has a very high error bar. Please check for accuracy.
5. Some of the figure legends needs more explanation.

Reviewer #2 (Remarks to the Author):

In this manuscript, Fels et al. attempt to develop a new derivative of methyl-oxalylglycine (MOG), with increased stability and an improved pharmacokinetic profile in order to study the biology of MCT2 and α -ketoglutarate (α -KG) in vivo. Based on their previous work showing that dimethyl-oxalylglycine (DMOG) is rapidly converted to MOG, the authors first show that MOG has a transient presence as it is de-esterified rapidly to from NOG in mouse blood. They then perform a series of in vitro structural-activity relationship (SAR) studies and functional evaluation on cell proliferation and apoptosis on a MCT2-overexpressing HCC1569 Her2+ breast cancer cell line in vitro, leading to IPOG as the primary candidate. These studies are complemented by in vivo orthotopic mouse tumor xenografts which indicated accumulation of NOG. The highlight of IPOG for a potential scaffold that could be built upon for future studies is interesting. Unfortunately, its effects on molecular regulation of α -KG and MCT2 expression are not evaluated in tumor tissues. Although IPOG has a little impact on tumor growth, evaluation of its association with altered hypoxic markers (HIF1a and PHD), deemed essential, is missing.

All in all, the consensus is that the manuscript does not reach the level of priority for publication in its present form.

Major concern(s):

- One open question that arises from this study; why is a breast cancer the only model utilized in this study, considering the universal role of α -KG/glutamine in regulating tumour growth? This is an important question as it will determine the role for MCT2 as a type of tumour specific. Unlike their previous work which focused primarily in tumorigenic metabolism, this is a drug development study. Also, one would expect a set data evaluating effects of IPOG on MCF7 (naturally high in MCT2 expression) parallel with enforced overexpression of MCT2 in HCC1569.

- Two of the most significant claims of this manuscript is that IPOG is able to "replicate the effects of MOG" and maintaining "a subset of metabolic effects of MOG". Unfortunately, its anti-tumour cell proliferation and apoptosis efficacy in vitro is not as competent as MOG. While the authors fairly convincingly show that IPOG has improved in vivo stability and pharmacokinetics that resulted in higher level of [NOG]IC in mice tumors, this does not formally demonstrate in vivo recapitulation of IPOG in vitro inhibitory effects. To better tie the data to the theme of the manuscript, the same (e.g., stabilization of HIF1a, tumor volume, etc.) should be shown in the tumors at similar (4 h) or longer timepoints.

- The authors hypothesized that the instability of MOG is due to high level of esterase activity in blood and thus compound 4-6 (branched esters) are derived to counter specificity of cellular esterases. However, these candidates were eliminated after in vitro experiments due to failure to meet two-fold cut off criterion, in the absence of the selective pressure – cellular esterases. Hence, one may envision that under in vivo (blood) selection, compounds 4-6 may be more resistant to degradation and thus more effective compared to other candidates such that the fold differences observed in vitro may be well extended.

- Despite the increased accumulation of NOG and succinate/malate in the tumor cells, IPOG has little impact on tumor growth in vivo. Given that α -KG is 'a key metabolic node', with a profound impact on physiology and pathophysiology, such as tumorigenesis, one would expect a significant inhibition of tumor growth by IPOG as compared to MOG. Albeit there is not biological effect of IPOG is not sufficient to alter overall tumor growth, it will likely alter tumor proliferation (Ki67), PHD/HIF1a or angiogenesis markers (CD31 or endomycin), at the minimum. These evaluations will establish a link between IPOG-induced NOG accumulation to tumor biology and increase the appeal of this study to the field, in particular on the α -KG-related functions.

- The reviewer are not persuaded that the modification only led to a "trade-off between blood stability and MCT2-dependent uptake". The chemical alteration on IPOG has made IPOG rather ineffective albeit increase in NOG accumulation; (1) no change in tumor growth [likely to be due to reduced effectiveness as seen in in vitro proliferation assay that only at high doses that it can impair proliferation]. In fact, it is hard to imagine that further structural modifications of IPOG can yield high bioavailability and stronger downstream effects in pathological modelling. It is reasonable to assume that bulkier alkyl group, as in IPOG, minimally increase steric hindrance and thus improves chemical and metabolic stability. But this would also reduce its structural flexibility needed to be de-esterification. Also, with any new modification, it is crucial to eliminate any potential interacting partners that arise from the modifications.

- Line 332: Evidence of PHD engagement by compound 2 and 3, although there was the responsive increased HIF1a stabilization.

- Need an explanation for lesser induction of HIF1a by compound 2 and 3 than MOG. The blots require a loading control, such as β -actin (Figure 3).

Minor concern:

- Line 163: compound 2 and 3 instead of compounds 2 and 3.

- Figure 3: The results of the quantification with how many repeats and statistical power calculation was not detailed in the paragraphs or the figure legends.

Reviewer #3 (Remarks to the Author):

Fets et al. "Development of novel MOG analogues with increased stability to explore MCT2 and α -ketoglutarate biology in vivo"

Fets et al. report on the development of novel analogues of MOG with increased stability in vivo to study α -ketoglutarate (α KG)-dependent processes. This works follows their recent finding (Fets et al. Nat. Chem. Biol. 2018) that DMOG, a tool compound used to study hypoxia, is rapidly hydrolysed in cell culture media into MOG, which is a substrate for the transporter MCT2. Intracellular MOG is then converted into the α KG analogue NOG. Importantly, in the same paper the authors reports that NOG do not only trigger hypoxia responses by targeting PHDs, but also affects glutamine metabolism by inhibiting GDH and IDH.

After observing that DMOG and MOG are unstable in mouse blood, the authors generated a series of 10 analogues, two of which show some NOG generation and activity in cell culture and, for one of these (IPOG), also increased stability in vivo.

Overall, I find the aim of the study of interest and the experimental part well conducted. Albeit limited by the relatively low number of analogues generated, the authors provide some initial information about the SAR between MCT2 and its substrates. My main concern is whether the modest increased stability of IPOG is sufficient to have a significant biological impact, and therefore support the usefulness of this analogue for future in vitro and in vivo studies. The two analogs (2 and 3/IPOG) retaining NOG-generation capacity are much less potent than MOG in all cell culture assay (Fig. 3) and, in vivo, the short-term increase in [NOG]_{ic} observed with IPOG vs. MOG does not seem to be sufficient to induce significant biological effects (Fig. 5).

Points to be addressed (my comments focus on the biological aspects of this manuscript and do not address chemistry, as this is not my core expertise):

1. The two NOG-generating analogues 2 and 3/IPOG show significant reduction in cellular uptake and on intracellular NOG concentration when compared to the MOG (Fig. 2). In line with this, 2 and 3/IPOG are clearly less active than MOG in all cell culture assay reported in Fig. 3, both short term (4h in Fig. 3c, e, f) or long term (Fig. 3a cell proliferation/viability over 96h). What is the stability of analogue 2 and 3/IPOG in cell culture media? If the analogues 2 and 3/IPOG are more stable, can they sustain HIF1 α stabilisation for longer time than MOG (beyond the 4h shown in Fig. 3c)?
2. All the data presented are derived from a single isogenic cell model (HCC1569 EV/MCT2) based on overexpression of MCT2. It would be important to assess the relative activity of the analogues compared to MOG in cell lines expressing endogenously MCT2 (HCC70, MCF7,...) that the authors used in their previous publication (Fets et al. Nat. Chem. Biol. 2018).
3. Does any of the analogues showing limited MCT2-mediated intracellular accumulation (4-7) act as an inhibitor of the transporter?
4. In the in vivo experiment of Fig. 4 and Fig. 5, the increased concentration of IPOG in blood is relatively transient (up to 1-1.5 h p.i.; Fig 4b), the increase of [NOG]_{ic} in tumor cells was measured only at 2h p.i. (Fig. 5c) and no significant biological effects were observed. I understand that in vivo experiments are demanding, but in my opinion the authors should provide additional evidences to support the added value of IPOG to study MCT2 and aKG biology in vivo, as they claim in the title.

Minor points:

- Figure 1e is not referenced in the text.
- Line 159-160: "...was associated with increased [NOG]_{IC} and increased apoptosis.." I did not find the data supporting the "increased [NOG]_{IC}" claim.
- Fig 3c: a WB loading control should be provided.

Reviewers' comments:

Referee #1: Cancer biology, metabolism, signaling pathways, miRNA, autophagy, apoptosis, biomarkers, tumor microenvironment, targeted therapies for cancer

Fets et al have demonstrated the effect of new MOG analogues in terms of stability and MCT2 biology.

However, some suggestions to improve this manuscript are:

1. Please improve the quality of Figure 3c.

> In the revised version of the manuscript (where this figure is now Fig. 4d) we have added loading controls. If the reviewer refers to the appearance of the HIF1 α bands, in our experience, HIF1 α bands with common commercially available antibodies vary in their migration patterns depending on tissue or cells of origin. The migration pattern of HIF1 α in HCC1569 shown in the figure in question is typical to other experiments we have performed in the lab, however, in new Fig. 4e with cultured MCF7 cells, the HIF1 α band is more compact and there is minimal background.

2. The source (company information) of secondary antibodies should be included.

> We apologise for this omission and agree this is important information. In the revised manuscript we have now included this information (lines 430-431)

3. Cell proliferation and apoptosis measurements (in materials and methods section) requires substantial elaboration.

> In the revised manuscript we have now re-written this section with more detail (lines 434-441).

4. Figure 4B has a very high error bar. Please check for accuracy.

> In the revised manuscript (where this figure is shown as Suppl. Fig. 2b), we now show the individual measurements that we inadvertently omitted in the original version. While we agree with the reviewer that these data are variable, from our experience with dosing mice with a range of unstable compounds, it is not uncommon to observe variability between animals, which we suspect is what the variation in these measurements represents.

5. Some of the figure legends needs more explanation.

> We have now reviewed and/or re-written the figure legends to ensure sufficient detail is provided to describe the data shown and include replicate numbers.

We thank the reviewer for their valuable suggestions, which helped improve our manuscript.

Referee #2: Metabolisms, tumour microenvironment, angiogenesis, cell signalling, oncogenes

In this manuscript, Fets et al. attempt to develop a new derivative of methyl-oxalylglycine (MOG), with increased stability and an improved pharmacokinetic profile in order to study the biology of MCT2 and α -ketoglutarate (α -KG) in vivo. Based on their previous work showing that dimethyl-oxalylglycine (DMOG) is rapidly converted to MOG, the authors first show that MOG has a transient presence as it is de-esterified rapidly to form NOG in mouse blood. They then perform a series of in vitro structural-activity relationship (SAR) studies and functional evaluation on cell proliferation and apoptosis on a MCT2-overexpressing HCC1569 Her2+ breast cancer cell line in vitro, leading to IPOG as the primary candidate. These studies are complemented by in vivo orthotopic mouse tumor xenografts which indicated accumulation of NOG. The highlight of IPOG for a potential scaffold that could be built upon for future studies is interesting. Unfortunately, its effects on molecular regulation of α -KG and MCT2 expression are not evaluated in tumor tissues. Although IPOG has a little impact on tumor growth, evaluation of its association with altered hypoxic markers (HIF1 α and PHD), deemed essential, is missing.

All in all, the consensus is that the manuscript does not reach the level of priority for publication in its present form.

> We thank the reviewer for their comments. As detailed in our response to the editor and, for the convenience of the reviewer, also in point 2.2 below, in our attempts to address the reviewer's comments, we reached the conclusion that neither the HCC1569 nor a new mouse xenograft model used for the revision experiments are suitable for testing the action of these analogues *in vivo*. This was primarily due to poor tumour vascularisation that contributed to low bioavailability of the compounds. Therefore, in our revised manuscript we have removed most of the *in vivo* data and associated claims and re-focused on the use of our analogues for cell culture studies.

Major concern(s):

2.1 One open question that arises from this study; why is a breast cancer the only model utilized in this study, considering the universal role of α -KG/glutamine in regulating tumour growth? This is an important question as it will determine the role for MCT2 as a type of tumour specific. Unlike their previous work which focused primarily in tumorigenic metabolism, this is a drug development study. Also, one would expect a set data evaluating effects of IPOG on MCF7 (naturally high in MCT2 expression) parallel with enforced overexpression of MCT2 in HCC1569.

> In our original manuscript, we focused on breast cancer because, from our previous work, we had extensive understanding of the metabolic effects of MOG in human breast cancer cell lines that would allow us to benchmark the effects of the analogues in these cells. Furthermore, our original intention was to test the analogues *in vivo*, and the mouse mammary gland is relatively more accessible for orthotopic growth of transplanted tumours compared to other organs.

However, we agree with the reviewer that the analogues we report should be tested in other cell lines that express endogenous MCT2 levels. In the revised manuscript, we present new data with three cell lines of various tissues of origin: MCF7 (human breast cancer), SN12C (human renal cell carcinoma) and LN229 (human glioblastoma). In new Fig. 3c, we show that while MOG is cytotoxic in all three lines, analogues **2** and **3** are not. We attribute this to the substantially lower levels of intracellular NOG concentration ($[NOG]_c$) that these analogues elicit in cells compared to MOG (new Fig. 3d), which are not sufficient to inhibit metabolic targets (new Figs. 4a and 4c). However, lower $[NOG]_c$ in cells treated with compounds **2** or **3** is still sufficient to inhibit PHDs, stabilise HIF1 α and increase the expression of prototypical HIF1 α target proteins LDHA and PKM2 after 24 h of treatment (new Fig. 4e). In summary, analogues **2** and **3** maintain the ability to inhibit PHDs, but, in contrast to MOG, they do not inhibit metabolism or cause cytotoxicity.

With these new insights in hand, we decided to use these analogues to deconvolute cellular effects of NOG mediated by PHDs from those mediated by its metabolic targets. Previous work by others (PMID:23085753) had suggested that glutamine metabolism provides α KG, a cofactor for PHDs, which were proposed to hydroxylate an as-yet unidentified protein that activates the amino acid sensor and critical cell growth regulator mTORC1. This evidence suggested a new mechanism

for the regulation of the prototypical amino acid sensor mTORC1 by glutamine that involved PHDs. Support for this model was, in part, provided by the observation that treatment of cells with DMOG inhibited mTORC1. In our revised manuscript, we confirmed that under conditions that MOG inhibits PHDs (shown by HIF1 α stabilisation) mTORC1 signalling is also inhibited (new Fig. 4e). However, although both analogues **2** and **3** inhibit PHDs, they do not inhibit mTORC1 after up to 8 h of treatment. Given that MOG inhibits glutaminolysis but the analogues do not, our new data clearly show that PHDs are not likely mediators of mTORC1 activation.

DMOG has been extensively used to inhibit prolyl hydroxylases (PHDs) and thereby stabilise HIF1 α to study its functions in cells. Several of the key HIF1 α gene targets mediate metabolic effects similar to those elicited by MOG in MCT2-expressing cells (e.g. inhibition of mitochondrial metabolism). Therefore, despite its popularity, (D)MOG is unsuitable for studies of the specific functions of HIF1 α because it is impossible to deconvolute its cellular effects from those of MOG. Our new data with cultured cells support the idea that MOG analogues are useful tools because they stabilise HIF1 α without eliciting the metabolic effects of MOG. We therefore feel that our new data justify a new focus of our manuscript on the use of MOG analogues as PHD inhibitors that lack the convoluting metabolic effects of MOG in cultured cells.

2.2 Two of the most significant claims of this manuscript is that IPOG is able to “replicate the effects of MOG” and maintaining “a subset of metabolic effects of MOG”. Unfortunately, its anti-tumour cell proliferation and apoptosis efficacy in vitro is not as competent as MOG. While the authors fairly convincingly show that IPOG has improved in vivo stability and pharmacokinetics that resulted in higher level of [NOG]IC in mice tumors, this does not formally demonstrate in vivo recapitulation of IPOG in vitro inhibitory effects. To better tie the data to the theme of the manuscript, the same (e.g., stabilization of HIF1 α , tumor volume, etc.) should be shown in the tumors at similar (4 h) or longer timepoints.

> We concede that the way we made these claims distracted from the useful attributes of the analogues in cultured cells while their *in vivo* functions required further experimental support.

As outlined in our response to the editor and also detailed here, for the reviewer's convenience, we performed several new experiments to address the functions of the analogues *in vivo*. To this end, we first considered the reasons why the effects of analogues were subdued in the xenograft tumours from HCC1569±MCT2 cells presented in our original manuscript. Macroscopic and histological analysis showed that these tumours are poorly vascularised, which could limit the amount of compounds that reached the tissue. Bearing in mind the reviewer's request for experiments in cells with endogenous MCT2, we generated xenograft tumours using SN12C, a human kidney cancer cell line that we have previously shown to express relatively high levels of MCT2 (Fets *et al.* 2018) and is sensitive to MOG. SN12C have been shown to form well-vascularised xenograft tumours (e.g. PMID: 20103651).

In Rev. Fig. 1A we confirm that MOG kills SN12C cells that express empty vector (SN12C-EV) and the MOG cytotoxicity is attenuated in SN12C cells where MCT2 has been knocked down (SN12C-shMCT2) confirming that these cells are a suitable model for the intended experiments. Both cell lines formed subcutaneous xenograft tumours that grew comparably in mice (Rev. Fig. 1B). Infusion of these mice with ¹³C-glutamine led to only a small fractional labelling of TCA intermediates that did not significantly change in tumours from mice treated with either MOG or compound **3** (not shown). We detected more NOG in SN12C-EV tumours from mice treated with MOG than those treated with **3** and although there was a trend of decreased NOG in SN12C-shMCT2 tumours, this was not significant (Rev. Fig. 1C). Our attempts with various antibodies to reliably detect HIF1 α were unsuccessful largely due to poor and variable HIF1 α signal from tissues (not shown). Finally, inspection of the tumours revealed that, despite our prediction, SN12C-derived tumours were also poorly vascularised. It is possible that these tumours could have developed a vascular system if they had been allowed to grow bigger, which, however, would have breached the humane end point limits allowed by our licence for animal work.

Review Fig. 1. New xenograft experiments to test MOG analogues *in vivo*.

A. Characterisation of SN12C cells that stably express empty vector (pLKO) or an shRNA to knock down MCT2 (shMCT2). Confluence and apoptosis were measured over time as in new Fig. 3c and show that SN12C-shMCT2 cells are less sensitive to MOG than SN12C-pLKO cells. For comparison, analogues **2** and **3** do not elicit cytotoxicity in either cell line. **B.** Growth of tumours from cells shown in (A) implanted subcutaneously in the flanks of NSG mice. **C.** Mice shown in (B) were administered with 100 mg/kg of either MOG or compound **3**. 2 h later, mice were killed, tumours rapidly removed and processed for metabolite extraction. Intratumoral NOG levels were determined by LC-MS analysis using a method similar to that described in the manuscript.

Given these limitations, we concluded that, to further investigate the *in vivo* effects of these analogues, we would need to screen various tumour models (e.g. with tumours grown orthotopically or that are better vascularised and, therefore, more likely to get exposed to the compounds) and optimise the enrichment of HIF1 α signal (e.g. through nuclear fractionation from fresh tissue), which, however, would require significantly more time. Therefore, in the revised manuscript we decided to remove most of our *in vivo* data and associated claims about the *in vivo* applications of these compounds and focus on their use in cultured cells as outlined in our response to point 2.1, above.

2.3 The authors hypothesized that the instability of MOG is due to high level of esterase activity in blood and thus compound 4-6 (branched esters) are derived to counter specificity of cellular esterases. However, these candidates were eliminated after *in vitro* experiments due to failure to meet two-fold cut off criterion, in the absence of the selective pressure – cellular esterases. Hence, one may envision that under *in vivo* (blood) selection, compounds 4-6 may be more resistant to degradation and thus more effective compared to other candidates such that the fold differences observed *in vitro* may be well extended.

> The reviewer raises the excellent possibility that the more stable analogues may be more likely to reach the target tissue. As we outline in point 2.2, above, a proper investigation of this question would require suitable mouse tumour models that would enable efficient delivery of the compounds to the tumours. As we were not able to obtain reliable data with our current *in vivo* models, we were not convinced that these models are suitable to pursue further investigations of the analogues *in vivo*, so we did not attempt to further address this question.

2.4 Despite the increased accumulation of NOG and succinate/malate in the tumour cells, IPOG has little impact on tumor growth *in vivo*. Given that α -KG is ‘a key metabolic node’, with a profound impact on physiology and pathophysiology, such as tumorigenesis, one would expect a significant inhibition of tumor growth by IPOG as compared to MOG. Albeit there is not biological effect of IPOG is not sufficient to alter overall tumor growth, it will likely alter tumor proliferation (Ki67), PHD/HIF1 α or angiogenesis markers (CD31 or endomycin), at the minimum. These evaluations will establish a link between IPOG-induced NOG accumulation to tumor biology and increase the appeal of this study to the field, in particular on the α -KG-related functions.

> We agree that evaluating proliferation and processes downstream of HIF1 α would be an excellent way to assess the functional effects of the analogues *in vivo*. Given the challenges in obtaining good bioavailability of the compounds in the two xenograft models we worked with, our failed attempts to obtain reliable HIF1 α signals in western blots, and re-focus of the manuscript on the cell culture applications of the compounds, we did not pursue this question any further.

2.5 The reviewer are not persuaded that the modification only led to a “trade-off between blood stability and MCT2-dependent uptake”. The chemical alteration on IPOG has made IPOG rather ineffective albeit increase in NOG accumulation; (1) no change in tumor growth [likely to be due to reduced effectiveness as seen in *in vitro* proliferation assay that only at high doses that it can impair proliferation]. In fact, it is hard to imagine that further structural modifications of IPOG can yield high bioavailability and stronger downstream effects in pathological modelling. It is reasonable to assume that bulkier alkyl group, as in IPOG, minimally increase steric hindrance and thus improves chemical and metabolic stability. But this would also reduce its structural flexibility needed to be de-esterification. Also, with any new modification, it is crucial to eliminate any potential interacting partners that arise from the modifications.

> We agree with the reviewer’s comments and in the revised manuscript we have removed the statement for a “trade-off between blood stability and MCT2-dependent uptake”. We have also re-written the corresponding parts of the discussion to reflect the new focus on the manuscript on the effects of the analogues in cultured cells, which we trust are well-justified by our new data.

2.6 Line 332: Evidence of PHD engagement by compound 2 and 3, although there was the responsive increased HIF1 α stabilization.

> The reviewer refers to the discussion of data that are now presented as Fig. 4d. In the revised manuscript, our new data (Fig. 4e) show that, in MCF7 cells, which express endogenous MCT2 levels, both compounds **2** and **3** stabilise HIF1 α to a degree and length of time that are comparable to that of MOG. In our response to point 2.7, below, we provide potential explanations about differences in the degree of HIF1 α stabilisation by MOG, **2** and **3** within the same cell line and between cell lines.

2.7 Need an explanation for lesser induction of HIF1 α by compound 2 and 3 than MOG. The blots require a loading control, such as β -actin (Figure 3).

> We apologise for the omission of loading controls, which we have now included in the revised version of this figure (new Fig. 4d). As we showed before (Fets *et al.* 2018, PMID: 30297875) and further support with new data in this manuscript (Figs. 3d, 4a and 4c), which proteins NOG targets depends on the concentration it reaches in cells. We are therefore tempted to suggest that differences in HIF1 α stabilisation by MOG, **2** and **3**, reflect the intracellular concentrations these compounds elicit, respectively, in cells: the less [NOG]_{ic}, the less PHDs are inhibited leading to less HIF1 α stabilisation (Fig. 3b and 4d in revised manuscript). On the other hand, the affinity of NOG for PHDs is in the low μ M range (e.g. PMID: 23234607); [NOG]_{ic} remains an order of magnitude higher than that in cells treated with any of the three compounds, so one would predict saturation of PHDs in all cases. These observations indicate that affinities of compounds for specific protein targets measured *in vitro* do not necessarily correspond to the actual affinities within the cellular milieu (discussed more extensively in our previous work PMID: 30297875, 31264961). This discrepancy may be e.g. because of differential compartmentalisation of the target protein and free compound, modification of the protein or engagement with binding partners that change the affinity for the compound.

Minor concern:

- Line 163: compound 2 and 3 instead of compounds 2 and 3.

> In the revised manuscript lines 189-191 we have now re-written this sentence.

- Figure 3: The results of the quantification with how many repeats and statistical power calculation was not detailed in the paragraphs or the figure legends.

> We apologise for the omission. In the revised manuscript, we have reviewed the legends and have now added this information where applicable.

We thank the reviewer for their valuable suggestions, which helped improve our manuscript.

Referee #3: Innate immunity, solute carriers, cell death**Fets et al. "Development of novel MOG analogues with increased stability to explore MCT2 and α -ketoglutarate biology in vivo"**

Fets et al. report on the development of novel analogues of MOG with increased stability in vivo to study α -ketoglutarate (aKG)-dependent processes. This work follows their recent finding (Fets et al. *Nat. Chem. Biol.* 2018) that DMOG, a tool compound used to study hypoxia, is rapidly hydrolysed in cell culture media into MOG, which is a substrate for the transporter MCT2. Intracellular MOG is then converted into the aKG analogue NOG. Importantly, in the same paper the authors reports that NOG do not only trigger hypoxia responses by targeting PHDs, but also affects glutamine metabolism by inhibiting GDH and IDH.

After observing that DMOG and MOG are unstable in mouse blood, the authors generated a series of 10 analogues, two of which show some NOG generation and activity in cell culture and, for one of these (IPOG), also increased stability in vivo.

Overall, I find the aim of the study of interest and the experimental part well conducted. Albeit limited by the relatively low number of analogues generated, the authors provide some initial information about the SAR between MCT2 and its substrates. My main concern is whether the modest increased stability of IPOG is sufficient to have a significant biological impact, and therefore support the usefulness of this analogue for future in vitro and in vivo studies. The two analogs (2 and 3/IPOG) retaining NOG-generation capacity are much less potent than MOG in all cell culture assay (Fig. 3) and, in vivo, the short-term increase in [NOG]_{ic} observed with IPOG vs. MOG does not seem to be sufficient to induce significant biological effects (Fig. 5).

> We thank the reviewer for their overall encouraging assessment of our study. We concede that the significantly more data would be required to solidify our claims about the use of the reported analogues *in vivo*. As outlined in our response to the editor's comments, above, and detailed in our response to the reviewer's specific points below, we attempted to provide more data on the function of these compounds *in vivo*. In short, we concluded that neither the orthotopic HCC1569, nor the new xenograft models we used (subcutaneously injected kidney cancer cells), are vascularised enough to allow sufficient bioavailability of MOG or the analogues to tumours. On the other hand, our new experiments provide substantial evidence that the analogues retain ability to inhibit PHDs but do not elicit the metabolism-mediated cytotoxic effects that MOG does. We therefore removed most of the *in vivo* data and associated claims, and re-focused our manuscript on the value of these compounds as tools to study PHDs without the convoluting effects of MOG on metabolism and cytotoxicity.

Points to be addressed (my comments focus on the biological aspects of this manuscript and do not address chemistry, as this is not my core expertise):

1. The two NOG-generating analogues 2 and 3/IPOG show significant reduction in cellular uptake and on intracellular NOG concentration when compared to the MOG (Fig. 2). In line with this, 2 and 3/IPOG are clearly less active than MOG in all cell culture assay reported in Fig. 3, both short term (4h in Fig. 3c, e, f) or long term (Fig. 3a cell proliferation/viability over 96h). What is the stability of analogue 2 and 3/IPOG in cell culture media? If the analogues 2 and 3/IPOG are more stable, can they sustain HIF1 α stabilisation for longer time than MOG (beyond the 4h shown in Fig. 3c)?

> We thank the reviewer for this insightful comment. In short, both analogues **2** and **3** are more stable than MOG in tissue culture media, however, this doesn't lead to more HIF1 α stabilisation even after 24 h of treatment.

In more detail, in new Fig. 2f, we show that both compounds **2** and **3** are more stable in tissue culture media than MOG. In new Fig. 4e, we show that analogues **2** and **3** stabilise HIF1 α with comparable kinetics to MOG. This is not surprising as all three compounds lead to mM-levels of [NOG]_{ic} that are expected to be sufficient to inhibit PHDs which bind NOG with low μ M affinity (PMID: 23234607). However, as we detail in our response to point 2 below, in contrast to MOG, analogues **2** and **3** do not inhibit glutamine metabolism and fail to inhibit mTORC1 signalling. We

therefore propose that these analogues are useful tools to study PHDs in cultured cells as they would allow researchers to deconvolute cellular effects that arise due to PHD inhibition from those due to inhibition of glutamine metabolism.

2. All the data presented are derived from a single isogenic cell model (HCC1569 EV/MCT2) based on overexpression of MCT2. It would be important to assess the relative activity of the analogues compared to MOG in cell lines expressing endogenously MCT2 (HCC70, MCF7,...) that the authors used in their previous publication (Fets *et al.* *Nat. Chem. Biol.* 2018).

> We agree that more data with cell lines that express endogenous MCT2 were needed. In the revised manuscript, we present new data with three human cancer cell lines of different tissue origins: MCF7 (breast cancer), SN12C (renal cell carcinoma) and LN229 (glioblastoma). In new Fig. 3c, we show that while MOG is cytotoxic in all three lines, analogues are not. We attribute this difference in cytotoxicity to the substantially lower levels of intracellular NOG concentration ($[NOG]_{IC}$) that these analogues elicit in cells compared to MOG (new Fig. 3d), which are not sufficient to inhibit metabolic targets (new Figs. 4a and 4c). However, lower $[NOG]_{IC}$ in cells treated with compounds **2** or **3** are still sufficient to inhibit PHDs, stabilise HIF1 α and increase the expression of prototypical HIF1 α targets LDHA and PKM2 after 24 h of treatment (new Fig. 4e). In summary, analogues **2** and **3** maintain the ability to inhibit PHDs, but, in contrast to MOG, they do not inhibit metabolism or cause cytotoxicity.

With these new insights in hand, we decided to use these analogues to deconvolute cellular effects of NOG mediated by PHDs from those mediated by its metabolic targets. Previous work by others (PMID:23085753) had suggested that glutamine metabolism provides α KG, a cofactor for PHDs, which were proposed to hydroxylate an as-yet unidentified protein that activates the amino acid sensor and critical cell growth regulator mTORC1. This evidence suggested a new mechanism for the regulation of the prototypical amino acid sensor mTORC1 by glutamine that involved PHDs. Support for this model was, in part, provided by the observation that treatment of cells with DMOG inhibited mTORC1. In our revised manuscript, we confirmed that under conditions that MOG inhibits PHDs (shown by HIF1 α stabilisation) mTORC1 signalling is also inhibited (new Fig. 4e). However, although both analogues **2** and **3** inhibit PHDs, they do not inhibit mTORC1 after up to 8 h of treatment. Given that MOG inhibits glutaminolysis but the analogues do not, our new data clearly show that PHDs are not likely mediators of mTORC1 activation.

DMOG has been extensively used to inhibit prolyl hydroxylases (PHDs) and thereby stabilise HIF1 α to study its functions in cells. Several of the key HIF1 α gene targets mediate metabolic effects similar to those elicited by MOG in MCT2-expressing cells (e.g. inhibition of mitochondrial metabolism). Therefore, despite its popularity, (D)MOG is unsuitable for studies of the specific functions of HIF1 α because it is impossible to deconvolute its cellular effects from those of MOG. Our new data with cultured cells support the idea that MOG analogues are useful tools because they stabilise HIF1 α without eliciting the metabolic effects of MOG. We therefore feel that our new data justify a new focus of our manuscript on the use of MOG analogues as PHD inhibitors that lack the convoluting metabolic effects of MOG in cultured cells.

3. Does any of the analogues showing limited MCT2-mediated intracellular accumulation (4-7) act as an inhibitor of the transporter?

> We thank the reviewer for this excellent hypothesis. In new Fig. 2e we provide evidence that, indeed, the ketone analogue (**7**) acts as an inhibitor. For assessing potential inhibitors of MCT2, we chose not to use the endogenous MCT2 substrate pyruvate because it is rapidly metabolised after entering cells, and would be challenging to infer MCT2 activity from changes in intracellular abundance of pyruvate. Given that MOG is an MCT2 substrate, we used the known ability of MOG to inhibit respiration as a readout of MCT2 inhibition: an MCT2 inhibitor should prevent MOG entry and would attenuate the ability of MOG to inhibit respiration. This approach also enabled the rapid screen of several compounds. In new Fig. 2e, we validate this approach by showing that the commercially available MCT2 inhibitor AR-C155858 prevents MOG from inhibiting respiration. We then tested analogues 4-7 and found that only **7** is able to prevent MOG from inhibiting respiration in intact cells. In the revised manuscript, we discuss the method in lines 443-453 and the new results in lines 153-166.

4. In the *in vivo* experiment of Fig. 4 and Fig. 5, the increased concentration of IPOG in blood is relatively transient (up to 1-1.5 h p.i.; Fig 4b), the increase of [NOG]_{ic} in tumor cells was measured only at 2h p.i. (Fig. 5c) and no significant biological effects were observed. I understand that *in vivo* experiments are demanding, but in my opinion the authors should provide additional evidences to support the added value of IPOG to study MCT2 and aKG biology *in vivo*, as they claim in the title.

> We agree with the reviewer that further evidence would be needed to ascertain the value of our MOG analogues for *in vivo* studies.

As outlined in our response to the editor and also detailed here, for the reviewer's convenience, we performed several new experiments to address the functions of the analogues *in vivo*. To this end, we first considered the reasons why the effects of analogues were subdued in the xenograft tumours from HCC1569±MCT2 cells presented in our original manuscript. Macroscopic and histological analysis showed that these tumours are poorly vascularised, which could limit the amount of compounds that reached the tissue. Bearing in mind the reviewer's request for experiments in cells with endogenous MCT2, we generated xenograft tumours using SN12C, a human kidney cancer cell line that we have previously shown to express relatively high levels of MCT2 (Fets *et al.* 2018) and is sensitive to MOG. SN12C have been shown to form well-vascularised xenograft tumours (e.g. PMID: 20103651).

In Rev. Fig. 1A we confirm that MOG kills SN12C cells that express empty vector (SN12C-EV) and the MOG cytotoxicity is attenuated in SN12C cells where MCT2 has been knocked down (SN12C-shMCT2) confirming that these cells are a suitable model for the intended experiments. Both cell lines formed subcutaneous xenograft tumours that grew comparably in mice (Rev. Fig. 1B). Infusion of these mice with ¹³C-glutamine led to only a small fractional labelling of TCA intermediates that did not significantly change in tumours from mice treated with either MOG or compound **3** (not shown). We detected more NOG in SN12C-EV tumours from mice treated with MOG than those treated with **3** and although there was a trend of decreased NOG in SN12C-shMCT2 tumours, this was not significant (Rev. Fig. 1C). Our attempts with various antibodies to reliably detect HIF1 α were unsuccessful largely due to poor and variable HIF1 α signal from tissues (not shown). Finally, inspection of the tumours revealed that, despite our prediction, SN12C-derived tumours were also poorly vascularised. It is possible that these tumours could have developed a vascular system if they had been allowed to grow bigger, which, however, would have breached the humane end point limits allowed by our licence for animal work.

Review Fig. 1. New xenograft experiments to test MOG analogues *in vivo*.

A. Characterisation of SN12C cells that stably express empty vector (pLKO) or an shRNA to knock down MCT2 (shMCT2). Confluence and apoptosis were measured over time as in new Fig. 3c and show that SN12C-shMCT2 cells are less sensitive to MOG than SN12C-pLKO cells. For comparison, analogues **2** and **3** do not elicit cytotoxicity in either cell line.

B. Growth of tumours from cells shown in (A) implanted subcutaneously in the flanks of NSG mice.

C. Mice shown in (B) were administered with 100 mg/kg of either MOG or compound **3**. 2 h later, mice were killed, tumours rapidly removed and processed for metabolite extraction. Intratumoral NOG levels were determined by LC-MS analysis using a method similar to that described in the manuscript.

Given these limitations, we concluded that, to further investigate the *in vivo* effects of these analogues, we would need to screen various tumour models (e.g. with tumours grown orthotopically or that are better vascularised and, therefore, more likely to get exposed to the compounds) and optimise the enrichment of HIF1 α signal (e.g. through nuclear fractionation from fresh tissue), which, however, would require significantly more time. Therefore, in the revised manuscript we decided to remove most of our *in vivo* data and associated claims about the *in vivo* applications of these compounds and focus on their use in cultured cells as outlined in our response to point 2, above.

Minor points:

- Figure 1e is not referenced in the text.

> We apologise for this omission. In the revised manuscript, we now refer to the synthesis strategy shown in this figure panel in line 113.

- Line 159-160: “..was associated with increased [NOG]IC and increased apoptosis..” I did not find the data supporting the “increased [NOG]IC” claim.

> We thank the reviewer for pointing out this error. In the original manuscript we were referring to [NOG]_{IC} measurements from a dataset we used in Fig. 3b. However, we do not present the comparison of [NOG]_{IC} in HCC1569-EV vs HCC1569-MCT2 cells, that our original text was alluding to. In the revised manuscript (lines 185-186) we have now removed the reference to [NOG]_{IC}.

- Fig 3c: a WB loading control should be provided.

> We apologise for this omission. The corresponding Ponceau-stained membranes used for this western blot have now been added in the revised figure (new Fig. 4d).

We thank the reviewer for their valuable suggestions, which helped improve our manuscript.

Reviewers' comments:

Reviewer #1 (Remarks to the Author):

The revised manuscript looks better now.

Reviewer #2 (Remarks to the Author):

In this revised manuscript, Fels et al. have developed a series of derivatives of methyl-oxalylglycine (MOG) and evaluated their stabilities and pharmacokinetic profiles. They postulate that these compounds can be used to study MCT2 functions whilst minimising external noises from metabolism and cytotoxicity. They go on to demonstrate how these compounds recapitulates NOG as a functional tool to study MCT2 while detaching away from conventional NOG interference. Notably, they report that MOG analogues can stabilise HIF1 α , and by extension, inhibit PHDs. Most importantly, they do not significantly affect cell viability and MCT2-related metabolism.

Overall, experimental results complement with the new revised focus and aim of this manuscript. Major concerns are addressed. The effects of analogues are sufficiently captured in various lines, thus strengthening the core of this study. I agree with the authors on the biological utility of these analogues as a tool to perform in-vitro, isolated functional study of MCT-2 and appreciate that the authors concurred with the reviewers' concern on the in-vivo properties, as previously reported in their first manuscript. Lastly, I appreciate the useful attributes of this study and will support its publication once relevant justifications as in 2.3 has been made to the editors.

Comments on rebuttal and revised manuscript

>2.1: The authors have tested the analogues in various cell lines, as suggested by the reviewer and revamped the primary focus of this manuscript. The results obtained (Figure 3) have reinforced their claim that MOG analogues can mimic MOG as PHD inhibitors but lack the convoluting metabolic effects of MOG in cultured cells.

>2.2: I acknowledge the authors effort in addressing the reviewers' concern on their previous in-vivo claims on the MOG analogues. The authors have demonstrated in Rev. Fig. 1 the difficulty in investigating the in-vivo effects of these analogues, which led to the revamp on the focus of this paper as outlined in their response to 2.1.

>2.3: I acknowledge the authors' justification in 2.2 of not pursuing further investigations of analogues in-vivo. However, I find it unusual in what the authors presented in Figure 2E and their observation of "very little tolerance for the α -methyl substitutions (4-6)". Whilst compounds 4-6 failed to meet 2-fold cut off criterion, they are able to elicit comparable inhibition of respiration as MOG shown in Figure 2E. Two open question arises from this observation; (1) Is this indicative of how low an activity of MCT2 is needed to achieve what that is reflected in the pathway (Figure 2E), (2) The potency of the NOG analogues generated by compounds 4-6. Hence it will be worthwhile for the authors to address this possibility in text so as to justify their choice of compound 2 and 3.

Notably, I commend the authors as the merit of Figure 2E is that they have identified a potential inhibitor of MCT2 by a ketone group modification which switches the nature of the interaction with MCT2 from substrate to inhibitor.

>2.4: I acknowledge the authors' justification in 2.2 of not pursuing further investigations of analogues in-vivo.

>2.5: I acknowledge the authors' justification in 2.2 of not pursuing further investigations of analogues in-vivo.

>2.6: While the authors did not provide direct evidence of PHD inhibition, they have provided an extension of it by establishing a relational link between [NOG]IC and the level of HIF-1 α stabilisation, coupled with possible explanations.

>2.7: The authors have supplemented a Ponceau-stained membranes as loading controls. As aforementioned, they have suggested that the less [NOG]IC, the less PHDs are inhibited leading to less HIF1 α stabilisation.

Reviewer #3 (Remarks to the Author):

Fets et al. Novel MOG analogues to explore the MCT2 pharmacophore, α -ketoglutarate biology and cellular effects of N-oxalyglycine.

In the revised version of the manuscript, Fets and colleagues introduced some substantial changes to re-focus their study moving away from the potential use in vivo of the MOG analogs they generated, to focus on their use for in vitro studies aiming at inhibiting PHD without the observed metabolic and cytotoxic effects of MOG. Compared to MOG, the analogues 2 and 3 show reduced MCT2-dependent transport, leading to a reduced [NOG]ic and thereby attenuated metabolic effect, while retaining detectable inhibition of PHDs.

The authors addressed my initial concerns by either providing convincing new experimental data or by removing the in vivo part of their study. While the absence of in vivo assessment of the new analogues reduces the overall impact, I think that the revised study with its new focus remains of interest for the field and for the readers of Communications Biology.

The main remaining concern I have is related to the new data presented and the claims regarding PHDs and mTORC1 activity (such as abstract line 31-32 "we use these analogues to show that glutaminolysis-induced activation of mTORC1 can be uncoupled from PHD activity"). While I tend to agree with the authors on this point, their claims are based only on single experiment presented in Fig. 4e, which in my opinion is not sufficient to fully support author's conclusions. Indeed, amino acid-dependent regulation of mTORC1 is very complex, with multiple sensing/regulatory mechanisms involved depending on the experimental settings. Duran and colleagues (ref 43) used in their study different experimental conditions and investigated the role of PHD on mTORC1 activation mostly in the context of amino acid starvation/restimulation. I therefore think that the authors need to either provide additional evidences using experimental conditions much closer to the one of the original study by Duran et al (i.e amino acid starvation/restimulation, shorter kinetics,...) or, alternatively, rephrase their comments on this point to take into account the limited experimental evidences they provide to support their claims.

Of note, a recent study proposed that PHD1 controls mTORC1 in a hydroxylation-independent manner (D'Hulst, G. et al. PHD1 controls muscle mTORC1 in a hydroxylation-independent manner by stabilizing leucyl tRNA synthetase. Nat Commun11, 174 (2020). <https://doi.org/10.1038/s41467-019-13889-6>).

Lastly, the authors mention in the discussion aKG-dependent TET enzymes, but it was not clear to me whether the [NOG]ic induced by analogues 2 and 3 is sufficient to inhibit these enzymes or not. Can the authors either experimentally assess this point or at least explicitly comment on this hypothesis?

Reviewer #1 (Remarks to the Author):

The revised manuscript looks better now.

Reviewer #2 (Remarks to the Author):

In this revised manuscript, Fels et al. have developed a series of derivatives of methyl-oxalylglycine (MOG) and evaluated their stabilities and pharmacokinetic profiles. They postulate that these compounds can be used to study MCT2 functions whilst minimising external noises from metabolism and cytotoxicity. They go on to demonstrate how these compounds recapitulates NOG as a functional tool to study MCT2 while detaching away from conventional NOG interference. Notably, they report that MOG analogues can stabilise HIF1 α , and by extension, inhibit PHDs. Most importantly, they do not significantly affect cell viability and MCT2-related metabolism.

Overall, experimental results complement with the new revised focus and aim of this manuscript. Major concerns are addressed. The effects of analogues are sufficiently captured in various lines, thus strengthening the core of this study. I agree with the authors on the biological utility of these analogues as a tool to perform in-vitro, isolated functional study of MCT-2 and appreciate that the authors concurred with the reviewers' concern on the in-vivo properties, as previously reported in their first manuscript. Lastly, I appreciate the useful attributes of this study and will support its publication once relevant justifications as in 2.3 has been made to the editors.

Comments on rebuttal and revised manuscript

- 2.1: The authors have tested the analogues in various cell lines, as suggested by the reviewer and revamped the primary focus of this manuscript. The results obtained (Figure 3) have reinforced their claim that MOG analogues can mimic MOG as PHD inhibitors but lack the convoluting metabolic effects of MOG in cultured cells.

- 2.2: I acknowledge the authors effort in addressing the reviewers' concern on their previous in-vivo claims on the MOG analogues. The authors have demonstrated in Rev. Fig. 1 the difficulty in investigating the in-vivo effects of these analogues, which led to the revamp on the focus of this paper as outlined in their response to 2.1.

- 2.3: I acknowledge the authors' justification in 2.2 of not pursuing further investigations of analogues in-vivo. However, I find it unusual in what the authors presented in Figure 2E and their observation of "very little tolerance for the a-methyl substitutions (4-6)". Whilst compounds 4-6 failed to meet 2-fold cut off criterion, they are able to elicit comparable inhibition of respiration as MOG shown in Figure 2E. Two open question arises from this observation; (1) Is this indicative of how low an activity of MCT2 is needed to achieve what that is reflected in the pathway (Figure 2E), (2) The potency of the NOG analogues generated by compounds 4-6. Hence it will be worthwhile for the authors to address this possibility in text so as to justify their choice of compound 2 and 3.

> We would like to clarify that the results in Figure 2e do not indicate that compounds 4-6 have any effect on respiration; rather, this assay was designed to test the ability of analogues to *prevent* the inhibitory effects of MOG on respiration. In Fig. 2e we show that even though cells were pre-incubated with the tested compounds at a 4-fold molar excess compared to MOG (1 mM vs 0.25 mM, respectively), when added to cells on top of these compounds, MOG can inhibit respiration to the same extent as in the absence of the other analogues.

To further clarify this point, in the revised version of the manuscript, we added a new panel in Figure 2e, middle, where we show the raw respirometry data that the original panel on the right is based on. This new middle panel is also discussed in the revised text lines 164-165. These data show that none of the tested compounds alone can suppress respiration, consistent with the idea that they are poorly taken up by cells. We hope that the schematic on the left of Fig. 2e helps clarify the assay design, however, we are happy to consider further specific suggestions for improving the presentation of our data.

We thank the reviewer for the opportunity to clarify this important point.

Notably, I commend the authors as the merit of Figure 2E is that they have identified a potential inhibitor of MCT2 by a ketone group modification which switches the nature of the interaction with MCT2 from substrate to inhibitor.

- 2.4: I acknowledge the authors' justification in 2.2 of not pursuing further investigations of analogues in-vivo.

- 2.5: I acknowledge the authors' justification in 2.2 of not pursuing further investigations of analogues in-vivo.

- 2.6: While the authors did not provide direct evidence of PHD inhibition, they have provided an extension of it by establishing a relational link between [NOG]IC and the level of HIF-1 α stabilisation, coupled with possible explanations.

- 2.7: The authors have supplemented a Ponceau-stained membranes as loading controls. As aforementioned, they have suggested that the less [NOG]IC, the less PHDs are inhibited leading to less HIF1 α stabilisation.

> We thank the reviewer again for their constructive input, which helped improve our manuscript, and appreciate their acknowledgement of our efforts.

Reviewer #3 (Remarks to the Author):

Fets et al. Novel MOG analogues to explore the MCT2 pharmacophore, α -ketoglutarate biology and cellular effects of N-oxalylglycine.

In the revised version of the manuscript, Fets and colleagues introduced some substantial changes to re-focus their study moving away from the potential use in vivo of the MOG analogs they generated, to focus on their use for in vitro studies aiming at inhibiting PHD without the observed metabolic and cytotoxic effects of MOG. Compared to MOG, the analogues 2 and 3 show reduced MCT2-dependent transport, leading to a reduced [NOG]ic and thereby attenuated metabolic effect, while retaining detectable inhibition of PHDs.

The authors addressed my initial concerns by either providing convincing new experimental data or by removing the in vivo part of their study. While the absence of in vivo assessment of the new analogues reduces the overall impact, I think that the revised study with its new focus remains of interest for the field and for the readers of Communications Biology.

The main remaining concern I have is related to the new data presented and the claims regarding PDHs and mTORC1 activity (such as abstract line 31-32 "we use these analogues to show that glutaminolysis-induced activation of mTORC1 can be uncoupled from PHD activity"). While I tend to agree with the authors on this point, their claims are based only on single experiment presented in Fig. 4e, which in my opinion is not sufficient to fully support author's conclusions. Indeed, amino acid-dependent regulation of mTORC1 is very complex, with multiple sensing/regulatory mechanisms involved depending on the experimental settings. Duran and colleagues (ref 43) used in their study different experimental conditions and investigated the role of PHD on mTORC1 activation mostly in the context of amino acid starvation/restimulation. I therefore think that the authors need to either provides additional evidences using experimental conditions much closer to the one of the original study by Duran et al (i.e amino acid starvation/restimulation, shorter kinetics,...) or, alternatively, rephrase their comments on this point to take into account the limited experimental evidences they provides to support their claims.

Of note, a recent study proposed that PHD1 controls mTORC1 in a hydroxylation-independent manner (D'Hulst, G. et al. PHD1 controls muscle mTORC1 in a hydroxylation-independent manner by stabilizing leucyl tRNA synthetase. Nat Commun11, 174 (2020). <https://doi.org/10.1038/s41467-019-13889-6>).

> We thank the reviewer for their constructive feedback for improving the way we present this part of our work. In retrospect, we agree that we should have done a better job in highlighting potential caveats in the interpretation of our data. We also thank the reviewer for reminding us of the relevant study by D'Hulst *et al.*

We have attempted to further support our thesis that PHD activity is dispensable for mTORC1 regulation. In new Supplementary Fig. 3d (discussed in new text lines 262-264) we show that the specific PHD inhibitor FG4592 stabilises HIF1 α (evidence that it successfully inhibited PHDs) but did not prevent mTORC1 activation (shown by phosphorylation of the mTORC1 substrate S6K at Thr389) after restimulation of starved cells with amino acids. Our experimental conditions for this experiment are similar to those used by Duran *et al.* (PMID: 23085753, e.g. Fig. 6b-d: 2 h amino acid deprivation, and restimulation with amino acids for up to 1 h), albeit in a different cell line.

Although this new experiment further supports the idea that PHD catalytic activity is not required for mTORC1 regulation, we agree that, to definitively prove the generalisation of our conclusions, more experiments would be needed. However, further expansion into this point would diverge the focus of our study away from the usefulness of MOG analogues. With these considerations, together with the reviewer's suggestions, in mind, in the revised manuscript abstract (line 32), we added 'under our experimental conditions'. Also, in the discussion we added new text (lines 356-359) highlighting that our results should be considered in light of the specific experimental conditions we used and that, in other settings, a role for PHDs in mTORC1 regulation cannot be excluded. We now also reference the study by D'Hulst (line 357, new citation #51),

We hope that the revised text provides an objective view of our new data in the context of previously published studies.

Lastly, the authors mention in the discussion aKG-dependent TET enzymes, but it was not clear to me whether the [NOG]_{ic} induced by analogues 2 and 3 is sufficient to inhibit these enzymes or not. Can the authors either experimentally assess this point or at least explicitly comment on this hypothesis?

> We thank the reviewer for the opportunity to expand on this interesting topic. Our rationale for this statement was that the published IC₅₀ of TET enzymes for NOG is lower than that of metabolic targets (e.g. in the μ M range for TET2, PMID: 28647531). Therefore, one would expect that the low [NOG]_{ic} elicited by MOG in cells with low MCT2 expression, or by analogues in cells with high MCT2 expression, would be sufficient to inhibit TETs.

We did attempt to assess the effect of MOG analogues on TET activity according to a protocol published in PMID: 26751286, however, we were not able to obtain consistent results with MOG-treated cells as a positive control. As this is not our lab's core expertise, we decided not to further pursue optimisation of this assay and to only discuss this interesting possibility in the text. In the revised manuscript we added text to further emphasise that our comment is speculative (lines 368-369); we can completely remove this paragraph, if more suitable.

REVIEWERS' COMMENTS:

Reviewer #2 (Remarks to the Author):

The authors have clarified my questions.

Reviewer #3 (Remarks to the Author):

I am satisfied with the revised version of the manuscript and I support its publication.